# Provably Robust Temporal Difference Learning for Heavy-Tailed Rewards

**Semih Cayci**
Department of Mathematics
RWTH Aachen University
Aachen, Germany
cayci@mathc.rwth-aachen.de

**Atilla Eryilmaz**
Department of Electrical and Computer Engineering
The Ohio State University
Columbus, OH 43210
eryilmaz.2@osu.edu

## Abstract

In a broad class of reinforcement learning applications, stochastic rewards have heavy-tailed distributions, which lead to infinite second-order moments for stochastic (semi)gradients in policy evaluation and direct policy optimization. In such instances, the existing RL methods may fail miserably due to frequent statistical outliers. In this work, we establish that temporal difference (TD) learning with a dynamic gradient clipping mechanism, and correspondingly operated natural actor-critic (NAC), can be provably robustified against heavy-tailed reward distributions. It is shown in the framework of linear function approximation that a favorable tradeoff between bias and variability of the stochastic gradients can be achieved with this dynamic gradient clipping mechanism. In particular, we prove that robust versions of TD learning achieve sample complexities of order $\mathcal{O}(\varepsilon^{-\frac{1}{p}})$ and $\mathcal{O}(\varepsilon^{-1-\frac{1}{p}})$ with and without the full-rank assumption on the feature matrix, respectively, under heavy-tailed rewards with finite moments of order $(1 + p)$ for some $p \in (0, 1]$, both in expectation and with high probability. We show that a robust variant of NAC based on Robust TD learning achieves $\tilde{\mathcal{O}}(\varepsilon^{-4-\frac{2}{p}})$ sample complexity. We corroborate our theoretical results with numerical experiments.

## 1 Introduction

In this paper, we develop a framework for robust reinforcement learning in the presence of rewards with heavy-tailed distributions. Heavy-tailed phenomena, stemming from frequently observed statistical outliers, have been ubiquitous in decision-making applications under uncertainty. To name a few examples, waiting times in wireless communication networks [44, 24, 58], completion times of SAT solvers [14], numerous payoff quantities (e.g., stock prices, consumer signals) in economics and finance [22, 35, 38, 37] exhibit heavy-tailed behavior. An important characteristic of heavy-tailed random variables is the infinite order of higher moments, which stems from the frequently occurring outliers.

In reinforcement learning (RL), the goal is to maximize expected total reward in a Markov decision process (MDP) by continual interactions with the unknown and dynamic environment. Among policy optimization methods, Natural actor-critic (NAC) method and its variants [51, 30, 46, 17, 27, 5, 29] have become particularly prevalent due to their desirable stability and versatility characteristics, emanating from the use of temporal difference (TD) learning as the critic for the policy evaluation component of the NAC operation. The existing theoretical analyses for temporal difference learning [4, 54] and natural policy gradient/actor-critic methods [62, 1, 57] assume that the stochastic gradients have finite second-order moments, or even they are bounded. In particular, it is unknown whether natural actor-critic with function approximation is robust for stochastic rewards of heavy-tailed distributions with potentially infinite second-order moments. Furthermore, in practice, these methods

37th Conference on Neural Information Processing Systems (NeurIPS 2023).

are prone to non-vanishing and even increasing error under heavy-tailed reward distributions (see Example 1). This motivates us for the following fundamental question in this paper:

*Can temporal difference learning with function approximation be modified to provably achieve global optimality under stochastic rewards with heavy tails?*

We provide an affirmative answer to the above question by proposing a simple modification to the TD learning algorithm, which yields robustness against heavy tails. In particular, we show that incorporating a dynamic gradient clipping mechanism with a carefully-chosen sequence of clipping radii can provably robustify TD learning and NAC with linear function approximation, leading to global near-optimality even under stochastic rewards of infinite variance.

**Example 1** (Failure of TD learning under heavy-tailed reward)**.** In this example, we consider a randomly-generated discounted-reward Markov reward process[1] $(X_t, R_t)_t$ on a state space $\mathbb{X}$ with $|\mathbb{X}| = 64$ states, with the discount factor $\gamma = 0.9$ and the reward $R_t(X_t) = r(X_t) + N_t - \mathbb{E}[N_t]$ with $N_t \overset{iid}{\sim} \mathsf{Pareto}(1, 1.4)$ for any $t$. In order to predict the value function, we use (projected) TD learning (see [4]) with linear function approximation based on Gaussian features of dimension $d = 4$ and projection radius $\rho = 30$. The performance results are shown in Figure 1. Since $R_t$ is heavy-tailed

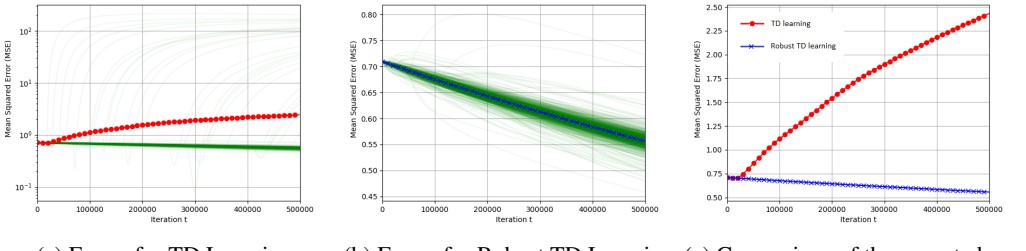

(a) Errors for TD Learning  (b) Errors for Robust TD Learning  (c) Comparison of the expected errors

Figure 1: Non-convergent behavior of TD learning under heavy-tailed noise with tail index 1.4. Each faded green line is the MSE for an individual trial, and the solid lines with markers indicate the average mean squared error for TD learning and Robust TD learning.

with infinite variance, the existing convergence results for traditional TD learning, which assume that $R_t$ has finite variance, do not hold. Furthermore, Figure 1 reveals that TD learning is prone to non-varnishing and even increasing error in practice despite the projection step, iterate averaging and small learning rate, due to the statistical outliers that cause extremely large error often as indicated by a non-negligible fraction of green lines in Figure 1a. On the other hand, with the same learning rate, projection radius and state-reward realizations, our robust variant of TD learning provides resilience against outliers (see Figure 1b), and leads to convergence in the expected behavior as in Figure 1c.

Stochastic rewards with heavy-tailed distributions appear in many important applications. Below, we briefly provide two motivating applications that necessitate robust RL methods to handle heavy tails.

**Application (1): Algorithm portfolios.** In solving complicated problems such as Boolean satisfiability (SAT) and complete search problems, which appear in numerous applications [19, 43], multiple algorithmic solutions with different characteristics are available. The algorithm selection problem is concerned about the minimization of total execution times to solve these problems [32, 49, 31], where different data distributions and machine characteristics, caused by recursive algorithms, are modeled as states, algorithm choices are modeled as actions, and the execution time of a selected algorithm is modeled as the cost (i.e., negative reward). It is well-known that the execution times, i.e., rewards, in the algorithm selection problem have *heavy-tailed distributions* with infinite-variance (e.g., $\mathsf{Pareto}(1, 1 + p)$ with $0 < p < 1$ as in [13]) similar to the case in Example 1 [13, 15, 49]. Thus, algorithm selection problem requires robust techniques that we consider here.

**Application (2): Scheduling for wireless networks.** The scheduling problem considers matching the users with random service demands to fading wireless channels (e.g., Gilbert-Elliot model) with stochastic transmission times so as to minimize the expected delay. A widely-adopted approach to

---

[1]The details of the setup, along with other numerical examples can be found in Section 4.

study the scheduling problem is to use MDPs (see, e.g., [40, 21, 10, 2]). It has been observed that the transmission times follow heavy-tailed distributions of infinite variance, due to various factors including the MAC protocol used, packet size, and channel fading [18, 58, 24, 23]. As such, solving this by using RL approach necessitates robust methods to handle heavy-tailed execution times.

**Main contributions.** Our main contributions in this paper contain the following.

• *Robust TD learning with dynamic clipping for heavy-tailed rewards.* We propose Robust TD learning with a dynamic gradient clipping mechanism, and prove that this TD learning variant with linear function approximation can achieve arbitrarily small estimation error that vanishes at rates $\mathcal{O}(T^{-\frac{p}{1+p}})$ and $\widetilde{\mathcal{O}}(T^{-p})$ without and with full-rank assumption on the feature matrix, respectively, even for heavy-tailed rewards with moments of order $1 + p$ for $p \in (0, 1]$. Our proof techniques make use of Lyapunov analysis coupled with martingale techniques for robust statistical estimation in dynamical systems, and can be of independent interest in the analysis of first-order methods.

• *Robust NAC under heavy-tailed rewards.* Based on Robust TD learning and the compatible function approximation result in [27], we propose a robust NAC variant, and show that $\mathcal{O}(\varepsilon^{-4-\frac{2}{p}})$ samples suffice to achieve $\varepsilon > 0$ error under standard concentrability coefficient assumptions.

• *High-probability error bounds.* We provide high-probability (sub-Gaussian) error bounds for the robust NAC and TD learning methods in addition to the traditional expectation bounds.

From a statistical viewpoint, our analysis in this work indicates a favorable bias-variance tradeoff: by introducing a vanishing bias to the semi-gradient via particular choices of dynamic gradient clipping, one can achieve *robustness* by eliminating the destructive impacts of statistical outliers even if the semi-gradient has infinite variance, leading to near optimality.

## 1.1 Related Work

**Temporal difference learning.** Temporal difference (TD) learning was proposed in [50], and has been the prominent policy evaluation method. The existing theoretical analyses of TD learning consider MDPs with bounded rewards [4, 7], or rewards with finite variance [54], while we consider heavy-tailed rewards. Our analysis utilizes the Lyapunov techniques in [4].

**Policy gradient methods.** Policy gradient (PG), and its variant natural policy gradient (NPG) have attracted significant attention in RL [59, 52, 27]. Recent theoretical works investigate the local and global convergence of these methods in the exact case, or with stochastic and bounded rewards [1, 57, 34, 39, 61]. As such, heavy-tailed rewards have not been considered in these works.

**Bandits with heavy-tailed rewards.** Stochastic bandit variants with heavy-tailed payoffs were studied in multiple works [6, 48, 33, 8]. The stochastic bandit setting can be interpreted as a very simple single-state (i.e., stateless or memoryless) model-based and tabular RL problem. The model we consider in this paper is a model-free RL setting on an MDP with a large state space, which is considerably more complicated than the bandit setting.

**Stochastic gradient descent with heavy-tailed noise.** There has been an increasing interest in the analysis of SGD with heavy-tailed gradient noise recently [56, 11, 16], following the seminal work of [45]. In our work, we consider the RL problem, which has significantly different dynamics than stochastic convex optimization.

**Robust mean and covariance estimation.** In basic statistical problems of mean and covariance estimation [42, 41, 36] and regression [20], the traditional methods do not yield the optimal convergence rates for heavy-tailed random variables, which led to the development of robust mean and covariance estimation techniques (for reference, see [36, 28]). Our paper utilizes tools from robust mean estimation literature (particularly, truncated mean estimator analysis in [6]), but considers the more complicated problem of TD learning and policy optimization in a dynamic environment rather than a static mean or covariance estimation problem with iid observations.

## 1.2 Notation

For a symmetric matrix $A \in \mathbb{R}^{d \times d}$, $\lambda_{\min}(A)$ denotes its minimum eigenvalue. $B_2(x, \rho) = \{y \in \mathbb{R}^d : \|x - y\|_2 \leq \rho\}$ and $\Pi_{\mathcal{C}}\{x\} = \arg\min_{y \in \mathcal{C}} \|x - y\|_2^2$ for any convex $\mathcal{C} \subset \mathbb{R}^d$.

## 2 Robust TD Learning for Value Prediction under Heavy Tails

First, we consider the problem of predicting the value function for a given discounted-reward Markov reward process with heavy-tailed rewards.

### 2.1 Value Prediction Problem

For a finite but arbitrarily large state space $\mathbb{X}$, let $(X_t)_{t\in\mathbb{N}}$ be an $\mathbb{X}$-valued Markov chain with the transition kernel $\mathcal{P} : \mathbb{X} \times \mathbb{X} \to [0,1]$. We consider a Markov reward process $(X_t, R_t)_{t\in\mathbb{N}}$ such that at state $X_t$, a stochastic reward $R_t = R_t(X_t)$ is obtained for all $t \geq 0$. For a discount factor $\gamma \in [0,1)$, the value function for the MRP $(X_t, R_t)_{t\in\mathbb{N}}$ is the following:

$$\mathcal{V}(x) = \mathbb{E}\Big[\sum_{t=1}^{\infty} \gamma^{t-1} R_t(X_t)\Big|X_1 = x\Big], \ x \in \mathbb{X}. \tag{1}$$

**Objective.** The goal is to learn $\mathcal{V}$ without knowing the transition kernel $\mathcal{P}$ by using samples from the system. In particular, for a parameterized class of functions $\{f_\Theta : \mathbb{X} \to \mathbb{R} : \Theta \in \mathbb{R}^d\}$, the goal is to solve the following stochastic optimization problem with mean squared error:

$$\min_{\Theta\in\mathbb{R}^d} \mathbb{E}_{x\sim\mu} |f_\Theta(x) - \mathcal{V}(x)|^2. \tag{2}$$

In order to solve (2) under Assumption 1, next we propose a robust variant of temporal difference (TD) learning with linear function approximation [50, 54].

### 2.2 Robust TD Learning Algorithm

For a given set of feature vectors $\{\Phi(x) \in \mathbb{R}^d : x \in \mathbb{X}\}$ with $\sup_{x\in\mathbb{X}} \|\Phi(x)\|_2 \leq 1$, we use $f_\Theta(\cdot) = \langle \Theta, \Phi(\cdot)\rangle$ as the approximation architecture. For a given dataset $\mathcal{D} = \{(X_t, R_t, X_t') \in \mathbb{X} \times \mathbb{R} \times \mathbb{X} : t \in \mathbb{N}\}$ with $X_t' \sim \mathcal{P}(X_t, \cdot)$, let the stochastic semi-gradient at $\Theta \in \mathbb{R}^d$ be defined as

$$g_t(\Theta) = \Big(R_t(X_t) + \gamma f_\Theta(X_t') - f_\Theta(X_t)\Big)\nabla_\Theta f_\Theta(X_t).$$

Robust TD learning is summarized in Algorithm 1.

---

**Algorithm 1:** Robust TD learning

---

**Inputs:** number of steps $T \geq 1$, clipping radii $(b_t)_{t\in[T]}$, projection radius $\rho > 0$, step-size $\eta > 0$
Set $\Theta(1) \in B_2(0,\rho)$                                                        `\\ initialization`
**for** $t = 1, 2, \ldots, T$ **do**

    $\Theta(t+1) = \Pi_{B_2(0,\rho)}\Big\{\Theta(t) + \eta_t \cdot g_t\big(\Theta(t)\big) \cdot \mathbb{1}\{\|g_t\big(\Theta(t)\big)\|_2 \leq b_t\}\Big\}$

**end for**
**Output:** $f_{\bar{\Theta}(T)}(\cdot) = \langle\bar{\Theta}(T), \Phi(\cdot)\rangle$ where $\bar{\Theta}(T) = \frac{1}{T}\sum_{t=1}^{T}\Theta(t)$

---

In the following, we will establish finite-time bounds for Robust TD learning by specifying the sequence of dynamic gradient clipping radii $(b_t)_{t\geq 1}$, projection radius $\rho$ and step-size $\eta$.

### 2.3 Finite-Time Bounds for Robust TD Learning

We make the following assumptions on the Markov reward process.

**Assumption 1.** The stochastic process $(X_t, R_t)_{t\in\mathbb{N}}$ satisfies the following:

1. Ergodicity: $(X_t)_{t\in\mathbb{N}}$ is an irreducible and aperiodic Markov chain with stationary distribution $\mu = \mu\mathcal{P}$. Also, we assume that there are constants $m > 0, \zeta \in (0,1)$ such that

$$\max_{x\in\mathbb{X}} \|\mathcal{P}^t(x,\cdot) - \mu\|_{\mathsf{TV}} \leq m\zeta^t, \ \forall t \in \mathbb{Z}_+. \tag{3}$$

2. Heavy-tailed reward: For some $p \in (0,1]$ and constant $u_0 \in (0,\infty)$,

$$\mathbb{E}[|R_t(X_t)|^{1+p}|X_t] \leq u_0 < \infty, \ a.s., \forall t \in \mathbb{N}. \tag{4}$$

3. Mean reward: For any $t \in \mathbb{N}$, $\mathbb{E}[R_t(X_t)|X_t] = r(X_t) \in [-1,1]$ a.s.

We note that the uniform ergodicity and bounded mean reward assumptions are standard in TD learning literature [3, 54, 4].

**Assumption 2** (Sampling). We consider two types of sampling strategies in this work:

    *2a. IID sampling:* $X_t \overset{iid}{\sim} \mu$ and $X_t' \sim \mathcal{P}(X_t, \cdot)$ for all $t \geq 1$.

    *2b. Markovian sampling:* $X_1 \sim \mu$ and $X_t' = X_{t+1} \sim \mathcal{P}(X_t, \cdot)$ for all $t \geq 1$.

**Assumption 3** (Realizability). There exists $\Theta^\star \in B_2(0, \rho)$ such that $\mathcal{V}(\cdot) = \langle \Theta^\star, \Phi(\cdot) \rangle$.

**Remark 1.** We note that Assumption 3 holds directly in interesting realizable problem classes, e.g., linear MDPs [25], and allows us to obtain results on the statistical error performance of our design. In cases when it does not hold, our results will continue to hold with an additional function approximation error proportional to $\inf_{\Theta \in B_2(0,\rho)} \sqrt{\mathbb{E}[|\mathcal{V}(x) - \langle \Theta, \Phi(x) \rangle|^2]}$, which is unavoidable due to the limitation of the linear function approximation architecture.

The following lemma is important in bounding the moments of the gradient norm under Robust TD learning in terms of the projection radius $\rho > 0$ and the upper bound $u_0$ on $\mathbb{E}[|R_t(X_t)|^{1+p}|X_t]$.

**Lemma 1** (Tail bounds for $\|g_t(\Theta(t))\|_2$). Let $\mathcal{F}_t^+ = \sigma(\Theta(1), \Theta(2), \ldots, \Theta(t), X_t)$ for $t \in \mathbb{Z}_+$. Then, under Assumption 1, we have:

$$\mathbb{E}[\|g_t(\Theta(t))\|_2^{1+p}|\mathcal{F}_t^+] \leq u < \infty, \ a.s., \tag{5}$$

for any $t \in \mathbb{Z}_+$, where $u = \min\{(u_0^{\frac{1}{1+p}} + 2\rho)^{1+p}, u_0 + 2^{2p+3}\rho^{1+p}\}$.

*Proof.* Note that we have $\mathbb{E}[\|g_t(\Theta(t))\|_2^{1+p}|\mathcal{F}_t^+] \leq \mathbb{E}[|R_t(X_t) + \gamma f_{\Theta(t)}(X_t') - f_{\Theta(t)}(X_t)|^{1+p}|\mathcal{F}_t^+]$ since $\sup_{x \in \mathbb{X}} \|\Phi(x)\|_2 \leq 1$. The upper bounds then follow by applying Minkowski's inequality and the triangle inequality for $L^p$ spaces, respectively, to this inequality. $\square$

This lemma will be useful in the analysis of both the expected (Theorems 1-2), and the high-probability bounds (Theorem 3) on the performance of Robust TD learning.

Next, we provide the main theoretical results in this paper: finite-time bounds for Robust TD learning. The proofs are mainly deferred to the appendix, while we provide a proof sketch for Theorem 3. In the following, we provide convergence bounds for the expected mean squared error under Robust TD learning with various choices of $b_t$.

**Theorem 1** (Expected error under Robust TD learning – iid sampling). Under Assumptions 1, 2a, 3, we have the following bounds for Robust TD learning:

**a)** For $b_t = (ut)^{\frac{1}{1+p}}$ for any $t \in \mathbb{Z}_+$ and $\eta_t = \eta = \frac{2\rho(1-\gamma)}{(uT)^{\frac{1}{1+p}}}$, we have:

$$\underset{\substack{\Theta(1), \Theta(2), \ldots, \Theta(T) \\ x \sim \mu}}{\mathbb{E}}\left[\left(\mathcal{V}(x) - \langle \bar{\Theta}(T), \Phi(x) \rangle\right)^2\right] \leq \frac{6\rho u^{\frac{1}{1+p}}}{(1-\gamma)T^{\frac{p}{1+p}}}, \ \forall T > 1. \tag{6}$$

**b)** Let $\Lambda = \sum_{x \in \mathbb{X}} \mu(x)\Phi(x)\Phi^\top(x)$, and $\mathfrak{C}_p(u, \lambda_{\min}, \gamma, \rho) = \frac{u}{1-\gamma}\left(4\rho + \frac{1}{(1-\gamma)\lambda_{\min}}\right)$. If $\lambda_{\min}(\Lambda) = \lambda_{\min} > 0$, then with the diminishing step-size $\eta_t = \frac{1}{(1-\gamma)t\lambda_{\min}}$ and $b_t = t$ for $t \in \mathbb{Z}_+$, for the average iterate $\bar{\Theta}(T)$, we have[2]:

$$\underset{\substack{\Theta(1), \Theta(2), \ldots, \Theta(T) \\ x \sim \mu}}{\mathbb{E}}\left(\mathcal{V}(x) - \langle \bar{\Theta}(T), \Phi(x) \rangle\right)^2 \leq \mathfrak{C}_p(u, \lambda_{\min}, \gamma, \rho)\left[\frac{\mathbf{1}_{p,1}T^{-p}}{1-p} + \frac{(1 - \mathbf{1}_{p,1})\log(eT)}{T}\right], \tag{7}$$

and for the last iterate $\Theta(T+1)$, we have:

$$\underset{\Theta(1), \ldots, \Theta(T)}{\mathbb{E}} \max_{x \in \mathbb{X}} |\mathcal{V}(x) - \langle \Phi(x), \Theta(T+1) \rangle|^2 \leq \frac{\mathfrak{C}_p(u, \lambda_{\min}, \gamma, \rho)}{\lambda_{\min}}\left[\frac{\log(eT)(1 - \mathbf{1}_{p,1})}{T} + \frac{\mathbf{1}_{p,1}T^{-p}}{1-p}\right], \tag{8}$$

for any $T > 1$, where $\mathbf{1}_{x,y} = 1$ if $x \neq y$ and 0 otherwise.

---

[2]The upper bound is $\mathfrak{C}_p(u, \lambda_{\min}, \gamma, \rho)\frac{1}{T}\sum_{t=1}^T t^{-p} = \widetilde{\mathcal{O}}(T^{-p})$, which is further upper bounded as (6) and (7) by using intergral bounds.

**Remark 2.** The convergence rates in Theorem 1 are $\mathcal{O}(T^{-\frac{p}{1+p}})$ and $\tilde{\mathcal{O}}(T^{-p})$ without and with the full-rank assumption $\lambda_{\min} > 0$, respectively. For $p = 1$, the convergence rates stated in Theorem 1 both match the existing results for TD learning with bounded rewards [4], up to a larger scaling factor of raw second-order moment rather than variance, due to clipping centered around 0.

**Remark 3** (Finite-time bounds in the unrealizable case). We note that our results hold without Assumption 3 as well. In this general case, there will be an additional function approximation error proportional to $\inf_{\Theta \in B_2(0,\rho)} \sqrt{\mathbb{E}[|\mathcal{V}(x) - \langle \Theta, \Phi(x) \rangle|^2]}$ (see Remark 1). One would use a richer function approximation scheme (e.g., larger projection radius $\rho$ or dimension $d$) to reduce this function approximation error, which will lead to an increase in the statistical error as we characterize in our bounds. For the extension of our analysis to the general case without Assumption 3, please see Appendix A.1.

In the following, we provide convergence bounds for Robust TD learning under Markovian sampling.

**Theorem 2** (Expected error under Robust TD learning – Markovian sampling). Under Assumptions 1,2b and 3, let $T > 1$, $\rho > 0$ be given, and define the mixing time $\tau = \min\{t \in \mathbb{Z}_+ : m\zeta^t \le \sqrt{2}\rho(uT)^{-\frac{1}{1+p}}\}$. Then, with $\eta_t = \eta = \sqrt{2}\rho(uT)^{-\frac{1}{1+p}}$, Robust TD learning yields the following:

$$\mathbb{E}_{\substack{\Theta(1),\Theta(2),\dots,\Theta(T) \\ x \sim \mu}} \left( \mathcal{V}(x) - \langle \bar{\Theta}(T), \Phi(x) \rangle \right)^2 \le \frac{7\rho u^{\frac{1}{1+p}}}{(1-\gamma)T^{\frac{p}{1+p}}} + \frac{2\sqrt{2}\rho(1+2\rho)(4\rho + \tau(1+6\rho))}{(1-\gamma)T^{\frac{1}{1+p}}}. \quad (9)$$

The proof of Theorem 2 is based on a similar Lyapunov technique as Theorem 1 in conjunction with the mixing time analysis in [4] for Markovian sampling, and can be found in Appendix A.

The bounds in Theorem 1 involve expectation over the parameters $\Theta(t), t \in [T]$. In the following, we provide a high-probability error bound on the mean squared error under Robust TD learning.

**Theorem 3** (High-probability bound for Robust TD learning). *For any $\delta \in (0,1)$, let $L_\delta = \log(4/\delta)$. Under Assumptions 1, 2a, 3, with step-size $\eta = \frac{\sqrt{2}(1-\gamma)\rho L_\delta^{\frac{1-p}{2(1+p)}}}{(uT)^{\frac{1}{1+p}}}$ and clipping radius $b_t = \left(\frac{ut}{L_\delta}\right)^{\frac{1}{1+p}}$,*

$$\sum_{x \in \mathbb{X}} \mu(x) \left( \mathcal{V}(x) - \langle \bar{\Theta}(T), \Phi(x) \rangle \right)^2 \le \frac{\rho u^{\frac{1}{1+p}}}{(1-\gamma)T^{\frac{p}{1+p}}} \left( 3L_\delta^{-\frac{1-p}{2(1+p)}} + 7L_\delta^{\frac{p}{1+p}} \right), \quad (10)$$

*holds with probability at least $1 - \delta$.*

In the following, we give a proof sketch for Theorem 3. The full proof can be found in Appendix A.

*Proof sketch.* Let $\mathcal{L}(\Theta) = \|\Theta - \Theta^\star\|_2^2$ be the Lyapunov function, and $\chi_t = 1 - \bar{\chi}_t = \mathbb{1}\{\|g_t\|_2 \le b_t\}$. Then, the Lyapunov drift can be decomposed as follows:

$$\mathcal{L}(\Theta(t+1)) - \mathcal{L}(\Theta(t)) \le 2\eta \mathbb{E}_t[g_t^\top(\Theta(t) - \Theta^\star)] + \eta^2 \mathbb{E}_t[\|g_t\|_2^2 \chi_t] + 2\eta B(t) + \eta^2 Z(t), \quad (11)$$

where

$$B(t) = g_t^\top(\Theta(t) - \Theta^\star)\chi_t - \mathbb{E}_t[g_t^\top(\Theta(t) - \Theta^\star)\chi_t] - \mathbb{E}_t[g_t^\top(\Theta(t) - \Theta^\star)\bar{\chi}_t],$$

is the bias in the stochastic semi-gradient, and

$$Z(t) = \|g_t\|_2^2 \chi_t - \mathbb{E}_t[\|g_t\|_2^2 \chi_t].$$

We can decompose $B(t)$ further into a martingale difference sequence

$$B_0(t) = g_t^\top(\Theta(t) - \Theta^\star)\chi_t - \mathbb{E}_t[g_t^\top(\Theta(t) - \Theta^\star)\chi_t],$$

and a bias term

$$B_\perp(t) = -\mathbb{E}_t[g_t^\top(\Theta(t) - \Theta^\star)\bar{\chi}_t].$$

By Freedman's inequality for martingales [12, 53], we have $\frac{1}{T}\sum_{t=1}^T B_0(t) \le \frac{7\rho u^{\frac{1}{1+p}} L_\delta^{\frac{p}{1+p}}}{T^{\frac{p}{1+p}}}$, and by Azuma inequality, we have $\frac{1}{T}\sum_{t=1}^t Z(t) \le \frac{u^{\frac{2}{1+p}} T^{\frac{1-p}{1+p}}}{L_\delta^{\frac{1-p}{1+p}}}$, each holding with probability at least $1 - \delta/2$.

By Hölder's inequality and Lemma 1, we can bound $B_\perp(t) \le ub_t^{-p}$ and $\mathbb{E}_t[\|g_t\|_2^2 \bar{\chi}_t] \le ub_t^{1-p}$, both with probability 1. Finally, by Lemma 2 in [54], we have the negative drift term

$$\mathbb{E}_t[g_t^\top(\Theta(t) - \Theta^\star)] \le -(1-\gamma) \sum_x \mu(x)(f_{\Theta(t)}(x) - \mathcal{V}(x))^2.$$

By telescoping sum of (11) and rearranging the terms, we have:

$$\frac{1}{T} \sum_{t=1}^T \|f_{\Theta(t)} - \mathcal{V}\|_\mu^2 \le \frac{\mathcal{L}(\Theta(1))}{2\eta(1-\gamma)T} + \frac{1}{(1-\gamma)T} \sum_{t=1}^T B(t) + \frac{\eta}{2(1-\gamma)T} \sum_{t=1}^T \Big(Z(t) + \mathbb{E}_t[\|g_t\|_2^2 \bar{\chi}_t]\Big).$$

The proof is concluded by substituting the above high probability bounds on the sample means of $B(t)$, $Z(t)$ and $\mathbb{E}_t[\|g_t\|_2^2 \bar{\chi}_t]$ (via union bound and integral upper bounds), and using Jensen's inequality on the left side of the above inequality. $\square$

Most notably, this important theorem establishes that, by appropriately controlling the bias term of dynamic gradient clipping to yield a vanishing sample mean with high probability as the number of iterations increases, one can limit the variance of the semi-gradient, thereby resulting in the provided global near-optimality guarantee.

# 3 Robust Natural Actor-Critic for Policy Optimization under Heavy Tails

In this section, we will study a two-timescale robust natural actor-critic algorithm (Robust NAC, in short) based on Robust TD learning, and provide finite-time bounds.

## 3.1 Policy Optimization Problem

We consider a discounted-reward Markov decision process (MDP) with a finite but arbitrarily large state space $\mathbb{S}$, finite action space $\mathbb{A}$, transition kernel $\mathcal{P}$ and discount factor $\gamma \in (0,1)$. The controlled Markov chain $\{(S_t, A_t) \in \mathbb{S} \times \mathbb{A} : t \in \mathbb{N}\}$ has the probability transition dynamics $\mathbb{P}(S_{t+1} \in s|S_1^t, A_1^t) = \mathcal{P}_{A_t}(S_t, s)$, for any $s \in \mathbb{S}$. Taking the action $A_t \in \mathbb{A}$ at state $S_t \in \mathbb{S}$ yields a random reward of $R_t(S_t, A_t)$ at any $t \in \mathbb{Z}_+$. For a given stationary randomized policy $\pi = (\pi(a|s))_{(s,a) \in \mathbb{S} \times \mathbb{A}}$, the value function $\mathcal{V}^\pi$ and the state-action value function (also known as Q-function) $\mathcal{Q}^\pi$ are defined as:

$$\mathcal{V}^\pi(s) = \mathbb{E}^\pi\Big[\sum_{t=1}^\infty \gamma^{t-1} R_t(S_t, A_t)\Big|S_1 = s\Big], \ s \in \mathbb{S} \tag{12}$$

$$\mathcal{Q}^\pi(s,a) = \mathbb{E}^\pi\Big[\sum_{t=1}^\infty \gamma^{t-1} R_t(S_t, A_t)\Big|S_1 = s, A_1 = a\Big], \ (s,a) \in \mathbb{S} \times \mathbb{A}. \tag{13}$$

**Remark 4 (From MDP to MRP).** Under any stationary randomized policy $\pi$, the process $(S_t, A_t)_{t>0} =: (X_t)_{t>0}$ is a Markov chain over the state-space $\mathbb{X} = \mathbb{S} \times \mathbb{A}$, thus $(X_t, R_t)$ with $R_t(X_t) = R_t(S_t, A_t)$ is a Markov reward process of the kind that we analyzed in Section 2. As such, we can use Robust TD learning to evaluate $\mathcal{V}(x) = \mathcal{Q}^\pi(x)$ for any $x = (s,a) \in \mathbb{S} \times \mathbb{A}$.

**Heavy-tailed reward.** We assume that the process $(X_t, R_t)_{t>0}$ with the Markov chain $X_t = (S_t, A_t)$ and the reward $R_t = R_t(X_t)$ satisfies Assumption 1. We denote the stationary distribution of $X_t = (S_t, A_t)$ under $\pi$ as $\mu^\pi$.

**Objective.** For an initial state distribution $\lambda$, the objective in this work is to find the following:

$$\pi^\star \in \arg\max_\pi \int_\mathbb{S} \mathcal{V}^\pi(s)\lambda(ds) =: \mathcal{V}^\pi(\lambda), \tag{14}$$

over the class of stationary randomized policies.

**Policy parameterization.** In this work, we consider a finite but arbitrarily large state space $\mathbb{S}$, and for such problems, the tabular methods do not scale [51, 3]. In order to address this scalability issue, we consider widely-used softmax parameterization with linear function approximation: for a given set of feature vectors $\{\Phi(s,a) \in \mathbb{R}^d : s \in \mathbb{S}, a \in \mathbb{A}\}$ and policy parameter $W \in \mathbb{R}^d$,

$$\pi_W(a|s) = \frac{\exp(W^\top \Phi(s,a))}{\sum_{a' \in \mathbb{A}} \exp(W^\top \Phi(s,a'))}, \ (s,a) \in \mathbb{S} \times \mathbb{A}. \tag{15}$$

In the following subsection, we will describe the robust natural actor-critic algorithm.

## 3.2 Robust Natural Actor-Critic Algorithm

For any iteration $k \geq 1$, we denote $\pi_k := \pi_{W(k)}$ throughout the policy optimization iterations.

For samples $\mathcal{D}^{(k)} = \{(S_{t,k}, A_{t,k}, R_{t,k}, S'_{t,k}, A'_{t,k}) : t \geq 1\}$, given $(b_{t,k})_{t,k \in \mathbb{Z}_+}$ and $\rho > 0$, Robust NAC Algorithm is summarized in Algorithm 2.

---

**Algorithm 2:** Robust Natural Actor-Critic Algorithm

---

    **Inputs:** clipping radii $(b_t)_{t \geq 1}$, projection radius $\rho > 0$, learning rate $\alpha > 0$, $L_\delta > 0$

    **for** $k = 1, 2, \ldots, K$ **do**

      Set $\Theta_k(1) = 0$ // `initialization: max-entropy policy`

      **for** $t = 1, 2, \ldots, T$ **do**

        Set $g_t^{(k)}(\Theta_k(t)) = \Big(R_{t,k} + \gamma f_{\Theta_k(t)}(S'_{t,k}, A'_{t,k}) - f_{\Theta_k(t)}(S_{t,k}, A_{t,k})\Big)\Phi(S_{t,k}, A_{t,k}).$

        $\Theta_k(t+1) = \Pi_{B_2(0,\rho)}\Big\{\Theta_k(t) + \eta_t \cdot g_t^{(k)}(\Theta_k(t)) \cdot \mathbb{1}\{\|g_t^{(k)}(\Theta_k(t))\|_2 \leq b_t\}\Big\}$

      **end for**

      $W(k+1) = W(k) + \alpha \cdot \frac{1}{T}\sum_{t=1}^T \Theta_k(t)$

    **end for**

---

**Remark 5.** The optimal solution $\Theta_k^\star \in \underset{\Theta \in \mathbb{R}^d}{\arg\min} \; \underset{x=(s,a)}{\mathbb{E}} \Big|\langle \Theta, \Phi(x)\rangle - \mathcal{Q}^{\pi_k}(x)\Big|^2$ is a good approximation of the natural policy gradient:

$$u_k = [G(\pi_k)]^{-1}\nabla_W \mathcal{V}^{\pi_k}(\lambda) \in \underset{w \in \mathbb{R}^d}{\arg\min} \; \underset{(s,a) \sim \mathsf{d}_\lambda^{\pi_k} \otimes \pi_k(\cdot|s)}{\mathbb{E}} \Big|\langle w, \nabla_W \log \pi_k(a|s)\rangle - \mathcal{A}^{\pi_k}(s,a)\Big|^2,$$

which follows from Jensen's inequality and leads to the Q-NPG [1]. For a detailed discussion, refer to Appendix B.

## 3.3 Finite-Time Bounds for Robust Natural Actor-Critic

In this subsection, we will provide finite-time bounds for Robust NAC.

We assume that the resulting Markov reward process under $\pi_k$ for each $k$ satisfies Assumptions 1-3 with stationary distribution $\mu^{\pi_k}$ and $u_k \geq \mathbb{E}[|R_{t,k}(S_{t,k}, A_{t,k})|^{1+p}|S_{t,k}, A_{t,k}]$. We assume that the dataset $\mathcal{D}^{(k)}$ is obtained independently at each iteration $k \geq 1$ for simplicity, with $(S_{t,k}, A_{t,k}) \overset{iid}{\sim} \mu^{\pi_k}$ and $S'_{t,k} \sim \mathcal{P}_{A_{t,k}}(S_{t,k}, \cdot)$ and $A_{t,k} \sim \pi_k(\cdot|S_{t,k})$ according to Assumption 2a under the stationary distribution $\mu^{\pi_k} = [\mu^{\pi_k}(s,a)]_{s \in \mathbb{S}, a \in \mathbb{A}}$ under $\pi_k$. We make the following standard assumption for policy optimization, which is common in the policy gradient literature [1, 57, 34].

**Assumption 4** (Concentrability). *For any $k \geq 1$, we assume that there exists $C_{\mathsf{conc}} < \infty$ such that:*

$$\max_{(s,a) \in \mathbb{S} \times \mathbb{A}} \frac{\mathsf{d}_\lambda^{\pi^\star}(s)\pi^\star(a|s)}{\mu^{\pi_k}(s,a)} \leq C_{\mathsf{conc}}, \tag{16}$$

*where $\mu^{\pi_k}$ is the stationary distribution of $(S_{t,k}, A_{t,k})_{t \geq 1}$ under $\pi_k$.*

**Theorem 4** (Finite-time bounds for Robust NAC). Under Assumptions 1-4 for any $k \geq 1$, for any $\delta \in (0,1)$ and $T, K > 1$, Robust NAC with $\rho \geq \max_k \|\Theta_k^\star\|_2$, $b_{t,k} = \left(\frac{u_k T}{\log(4T/\delta)}\right)^{\frac{1}{1+p}}$, learning rates $\eta = \frac{\sqrt{2}(1-\gamma)\rho L_\delta^{\frac{1-p}{2(1+p)}}}{(\max_{1 \leq k \leq K} u_k T)^{\frac{1}{1+p}}}$ and $\alpha = \frac{\sqrt{\log|\mathbb{A}|}}{\rho\sqrt{K}}$ achieves the following with probability at least $1 - \delta$:

$$\min_{1 \leq k \leq K}\{\mathcal{V}^{\pi^\star}(\lambda) - \mathcal{V}^{\pi_k}(\lambda)\} \leq \frac{2\rho\sqrt{\log|\mathbb{A}|}}{(1-\gamma)\sqrt{K}} + \sqrt{\frac{(\max_{1 \leq k \leq K} u_k)^{\frac{1}{1+p}}C_{\mathsf{conc}}\rho}{(1-\gamma)^3 T^{\frac{p}{1+p}}}\left(3L_\delta^{-\frac{1-p}{2(1+p)}} + 7L_\delta^{\frac{p}{1+p}}\right)},$$

where $L_\delta = \log(4T/\delta)$.

The proof of Theorem 4 can be found in Appendix B.

**Remark 6** (Sample complexity of Robust NAC). An immediate consequence of Theorem 4 is as follows: the best iterate error decays at a rate $\tilde{\mathcal{O}}(\frac{1}{\sqrt{K}}) + \tilde{\mathcal{O}}(\frac{1}{T^{\frac{p}{2(1+p)}}})$ after $K$ iterations of natural policy gradient, which contains $T$ steps of Robust TD learning per iteration. As such, in order to achieve $\varepsilon > 0$ error, one needs $T \times K = \tilde{\mathcal{O}}(\varepsilon^{-2-2(1+p)/p})$ samples.

**Remark 7.** Theorem 4 can be easily extended to expected error bounds and the full-rank case, where we would have $\tilde{\mathcal{O}}(K^{-1/2} + T^{-p})$ by using Theorem 1. By extending the analysis in Theorem 2, one can prove results for Markovian sampling as well.

**Remark 8** (Exploration in Robust NAC). Assumption 4 implies that the stationary state-action distribution under $\pi_k$ should be sufficiently exploratory so that it should have a nonzero probability at each state-action pair that is visited by the optimal policy $\pi^\star$. This assumption on the exploratory behavior of policy gradient methods is standard in reinforcement learning literature [1, 57, 34, 63]. Alternatively, entropy regularization can be used for natural policy gradient methods to encourage exploration, which would imply weaker conditions at the expense of an additional bias term in the function approximation setting [47, 9].

## 4  Numerical Results

In this section, we present numerical results for Robust TD learning and its non-robust counterpart.

**(1) Randomly-Generated MRP.** In the first example, we consider a randomly-generated MRP with $|\mathbb{X}| = 256$. The transition kernel is randomly generated such that $\mathcal{P}(x, x') \overset{iid}{\sim} \mathsf{Unif}(0, 1)$, and row-wise normalized to obtain a stochastic matrix. The feature dimension is $d = 128$, random features are generated according to the $\chi$-squared distribution $\Phi(x) = \Phi_0(x)/\|\Phi_0(x)\|_2$ with $\Phi_0(x) \sim \mathcal{N}(0, I_d)$ for all $x \in \mathbb{X}$, $\Theta^\star \sim 3U/\sqrt{d}$ for $U \sim \mathsf{Unif}^d(0, 1)$ and $\Psi = [\text{— } \Phi^\top(x) \text{ —}]_{x \in \mathbb{X}}$. The discount factor is $\gamma = 0.9$, and the reward is $R_t(X_t) = r(X_t) + N_t - \mathbb{E}[N_t]$ with $N_t \overset{iid}{\sim} \mathsf{Pareto}(1, 1.2)$. Mean squared error (2) under Robust TD learning and TD learning with the clipping radius $b_t = t$ and diminishing step-size $\eta_t = \frac{1}{\lambda_{\min}(1-\gamma)t}$ in Theorem 1 and projection radius $\rho = 30$ are shown in Figure 2. Despite

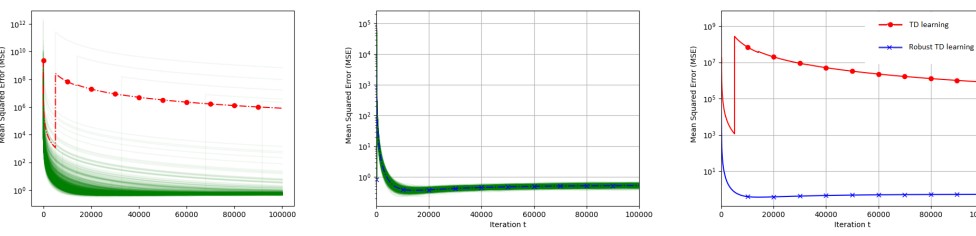

(a) Errors for TD Learning    (b) Errors for Robust TD Learning   (c) Comparison of the expected errors

Figure 2: Performance of TD learning and Robust TD learning under heavy-tailed rewards of tail index 1.2. Each faded green line is the MSE for an individual trial, and the solid lines with markers indicates the average error performance for TD learning and Robust TD learning.

diminishing step-size and projection, TD learning fails miserably often and in expectation due to the outliers in the reward that lead to extremely large errors (Figure 2a). On the other hand, for the same feature vectors, state and reward realizations, Robust TD learning effectively eliminates them in every sample path, and achieves good and consistent performance despite extremely heavy-tailed reward and gradient noise with tail index 1.2 (Figure 2a).

**(2) Circular Random Walk.** In this example, we consider a circular random walk for $\mathbb{X} = \{1, 2, \ldots, 256\}$, where each state $x$ is modulo-$|\mathbb{X}|$ [60]. The transition matrix is generated as $\mathcal{P}(x, x') = 1/3$ if $x = x'$ and $\mathcal{P}(x, x') = 1/24$ if $0 < |x - x'| \leq 8$. The reward and random feature generation is the same as the first example. The performances of TD learning and Robust TD learning in this structured case after 1000 trials are given in Figure 3.

A similar behavior as the randomly-generated MRP is observed in this example: due to outliers, TD learning fails miserably, while Robust TD learning achieves good performance consistently.

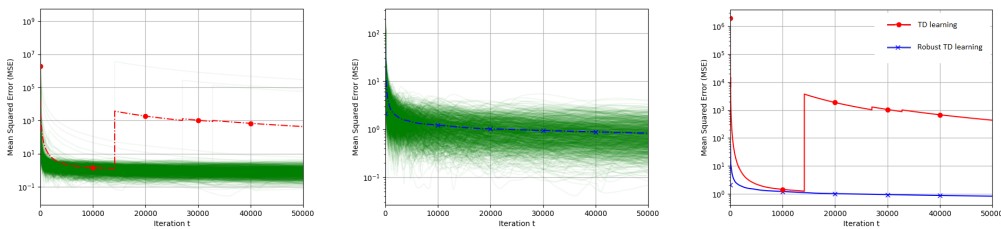

(a) Errors for TD Learning    (b) Errors for Robust TD Learning   (c) Comparison of the expected errors

Figure 3: Performances of Robust TD learning and TD learning for the circular random walk under heavy-tailed reward with tail index 1.2. Each faded green line is the error trajectory for an individual trial, and the solid lines indicate the expected errors for TD learning and Robust TD learning.

# 5 Conclusion

In this paper, we considered RL problem with heavy-tailed rewards, and proposed robust TD learning and NAC variants with a dynamic gradient clipping mechanism with provable performance guarantees, both in expectation and with high probability. Motivated by the results in this work, it would be interesting to explore single-timescale robust NAC and off-policy NAC for future work.

## Acknowledgments and Disclosure of Funding

Atilla Eryilmaz's research was supported in part by NSF AI Institute (AI-EDGE) 2112471, CNS-NeTS-2106679, CNS-NeTS-2007231; and the ONR Grant N00014-19-1-2621.

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

# A  Proofs for Robust TD Learning

The following lemma will be critical in our proofs.

**Lemma 2** (Lemma 4 in [54]). *For any two vectors $\widehat{V}, V \in \mathbb{R}^{|\mathbb{X}|}$,*

$$\|\mathcal{T}\widehat{V} - \mathcal{T}V\|_\mu \leq \gamma \cdot \|\widehat{V} - V\|_\mu,$$

*where*

$$(\mathcal{T}V)(x) = r(x) + \gamma \sum_{x' \in \mathbb{X}} \mathcal{P}(x, x')V(x'), \tag{17}$$

*is the Bellman operator.*

*Proof of Theorem. 1.* The proof follows the Lyapunov approach in [4]. Let $\mathcal{L}(\Theta) = \|\Theta - \Theta^*\|_2^2$ be the Lyapunov function for any $\Theta \in \mathbb{R}^d$. Then, by the non-expansivity of $\Pi_{\mathcal{B}_2(0,\rho)}$, we have:

$$\mathcal{L}(\Theta(t+1)) \leq \mathcal{L}(\Theta(t)) + \eta^2 \|g_t(\Theta(t))\|_2^2 \mathbb{1}\{\|g_t(\Theta(t))\|_2 \leq b_t\}$$
$$- 2\eta g_t(\Theta(t))^\top (\Theta(t) - \Theta^*) \mathbb{1}\{\|g_t(\Theta(t))\|_2 \leq b_t\}. \tag{18}$$

Taking conditional expectation given $\mathcal{F}_t$ and using the fact that $\mathbb{1}\{\|g_t(\Theta(t))\|_2 > b_t\} = 1 - \mathbb{1}\{\|g_t(\Theta(t))\|_2 \leq b_t\}$, we get:

$$\mathbb{E}[\mathcal{L}(\Theta(t+1))|\mathcal{F}_t] \leq \mathcal{L}(\Theta(t)) + 2\eta \mathbb{E}_t[g_t^\top(\Theta(t))(\Theta(t) - \Theta^\star)]$$
$$- 2\eta \mathbb{E}[g_t(\Theta(t))^\top(\Theta(t) - \Theta^*)\mathbb{1}\{\|g_t(\Theta(t))\|_2 > b_t\}|\mathcal{F}_t] + \eta^2 u b_t^{1-p}, \tag{19}$$

where we used

$$\mathbb{E}[\|g_t(\Theta(t))\|_2^2 \mathbb{1}\{\|g_t(\Theta(t))\|_2 \leq b_t\}|\mathcal{F}_t] \leq \mathbb{E}[\|g_t(\Theta(t))\|_2^{1+p} b_t^{1-p}|\mathcal{F}_t],$$
$$\leq u b_t^{1-p}, \tag{20}$$

in the last term. Now, for $\mathbb{E}_t[g_t^\top(\Theta(t))(\Theta(t) - \Theta^\star)]$, we have the following inequality:

$$\mathbb{E}_t[g_t^\top(\Theta(t))(\Theta(t) - \Theta^\star)] = \mathbb{E}_t[(R_t + \gamma f_{\Theta(t)}(X_t') - f_{\Theta(t)}(X_t))(f_{\Theta(t)}(X_t) - \mathcal{V}(X_t))],$$
$$= \mathbb{E}_t\Big[\Big((\mathcal{T}f_{\Theta(t)})(X_t) - f_{\Theta(t)}(X_t)\Big)\Big(f_{\Theta(t)}(X_t) - \mathcal{V}(X_t)\Big)\Big],$$

where $\mathcal{T}$ is the Bellman operator (17). By using the fact that the value function $\mathcal{V}$ is the fixed point of the Bellman operator $\mathcal{T}$, we have the following:

$$\mathbb{E}_t\Big[\Big((\mathcal{T}f_{\Theta(t)})(X_t) - f_{\Theta(t)}(X_t)\Big)\Big(f_{\Theta(t)}(X_t) - \mathcal{V}(X_t)\Big)\Big]$$
$$= \mathbb{E}_t\Big[\Big(\mathcal{T}f_{\Theta(t)}(X_t) - \mathcal{T}\mathcal{V}(X_t)\Big)\Big(f_{\Theta(t)}(X_t) - \mathcal{V}(X_t)\Big)\Big] - \mathbb{E}_t\Big[\Big(f_{\Theta(t)}(X_t) - \mathcal{V}(X_t)\Big)^2\Big]. \tag{21}$$

By using Lemma 2, we conclude that:

$$\mathbb{E}[g_t^\top(\Theta(t))(\Theta(t) - \Theta^\star)] \leq -(1-\gamma)\sum_{x \in \mathbb{X}} \mu(x)\Big(f_{\Theta(t)}(x) - \mathcal{V}(x)\Big)^2 = -(1-\gamma)\|f_{\Theta(t)} - \mathcal{V}\|_\mu^2. \tag{22}$$

Then, we can rewrite (19) as follows:

$$\mathbb{E}[\mathcal{L}(\Theta(t+1))|\mathcal{F}_t] \leq \mathcal{L}(\Theta(t)) - 2(1-\gamma)\eta\|f_{\Theta(t)} - \mathcal{V}\|_\mu^2$$
$$- 2\eta \mathbb{E}[g_t(\Theta(t))^\top(\Theta(t) - \Theta^*)\mathbb{1}\{\|g_t(\Theta(t))\|_2 > b_t\}|\mathcal{F}_t] + \eta^2 u b_t^{1-p}, \tag{23}$$

The bias introduced by using the gradient clipping can be bounded as follows:

$$\mathbb{E}[g_t(\Theta(t))^\top(\Theta(t) - \Theta^*)\mathbb{1}\{\|g_t(\Theta(t))\|_2 > b_t\}|\mathcal{F}_t]$$
$$\leq 2\rho\mathbb{E}[\|g_t(\Theta(t))\|_2 \mathbb{1}\{\|g_t(\Theta(t))\|_2 > b_t\}|\mathcal{F}_t], \tag{24}$$

which follows from Cauchy-Schwarz inequality, triangle inequality and the fact that $\max\{\|\Theta(t)\|_2, \|\Theta^*\|_2\} \leq \rho$ due to projection. Using Hölder's inequality on the RHS of (24), we obtain:

$$\mathbb{E}[g_t(\Theta(t))^\top(\Theta(t) - \Theta^*)\mathbb{1}\{\|g_t(\Theta(t))\|_2 > b_t\}|\mathcal{F}_t] \leq 2\rho u^{\frac{1}{1+p}}[\mathbb{P}(\|g_t(\Theta(t))\|_2 > b_t|\mathcal{F}_t)]^{\frac{p}{1+p}}.$$

Using Markov's inequality, we bound the bias due to using the clipped stochastic gradient as:

$$\mathbb{E}[g_t(\Theta(t))^\top(\Theta(t) - \Theta^*)\mathbb{1}\{\|g_t(\Theta(t))\|_2 > b_t\}|\mathcal{F}_t] \leq 2\rho u b_t^{-p}. \tag{25}$$

Substituting (25) into (19), and taking expectation over the trajectory $\mathcal{F}_t$, we obtain:

$$\mathbb{E}[\mathcal{L}(\Theta(t+1)) - \mathcal{L}(\Theta(t))] \leq -2\eta(1-\gamma)\|f_{\Theta(t)} - \mathcal{V}\|_\mu^2 + 4\eta\rho u b_t^{-p} + \eta^2 u b_t^{1-p}.$$

Telescoping sum over $t = 1, 2, \ldots, T$ yields:

$$\mathbb{E}\mathcal{L}(\Theta(T+1)) - \mathcal{L}(\Theta(1)) \leq -2\eta(1-\gamma)\sum_{t=1}^{T}\left(\mathbb{E}\|f_{\Theta(t)} - \mathcal{V}\|_\mu^2\right)$$
$$+ 4\eta\rho u \int_0^T b_s^{-p}ds + \eta^2 u \int_0^T b_s^{1-p}ds. \tag{26}$$

Rearranging the terms, using Jensen's inequality and $\mathcal{L}(\Theta(1)) \leq 4\rho^2$, and substituting the step-size $\eta$ yields the result.

(b) For the full-rank case, note that

$$\|f_\Theta - \mathcal{V}\|_\mu^2 = (\Theta - \Theta^\star)^\top\left(\sum_{x\in\mathbb{X}}\mu(x)\Phi(x)\Phi^\top(x)\right)(\Theta - \Theta^\star),$$
$$\geq \lambda_{\min}\|\Theta - \Theta^\star\|_2^2,$$

which implies (together with (23)) that:

$$\mathbb{E}\|\Theta(t+1) - \Theta^\star\|_2^2 \leq (1-\eta_t\lambda(1-\gamma))\|\Theta(t) - \Theta^\star\|_2^2 - \eta_t(1-\gamma)\mathbb{E}\|f_{\Theta(t)} - \mathcal{V}\|_\mu^2 + 4\eta_t\rho u b_t^{-p} + \eta_t^2 b_t^{1-p} u.$$

With the step-size choice $\eta_t = \frac{1}{(1-\gamma)\lambda t}$, we obtain by induction:

$$\mathbb{E}\|\Theta(t+1) - \Theta^\star\|_2^2 \leq -\frac{1}{\lambda t}\sum_{k=1}^{t}\mathbb{E}\|f_{\Theta(k)} - \mathcal{V}\|_\mu^2 + \frac{4\rho u}{\lambda_{\min}t}\sum_{k=1}^{t}b_k^{-p} + \frac{u}{\lambda_{\min}^2 t}\sum_{k=1}^{t}\frac{b_k^{1-p}}{k}.$$

By rearranging the terms and using the integral bound for the summations above, and using the Jensen's inequality for the $\mu$-norm, we obtain the result. $\qquad\square$

*Proof of Theorem 2.* Let $\mathcal{F}_t^{++} = \sigma(\Theta(1), \ldots, \Theta(t), X_t, X_{t+1})$ and $\mathbb{E}_t^{++}[\cdot] = \mathbb{E}[\cdot|\mathcal{F}_t^{++}]$. Also, let

$$\hat{g}(\Theta) = \mathbb{E}_t^{++}g_t(\Theta),$$
$$\bar{g}(\Theta) = \sum_{x,x'\in\mathbb{X}}\mu(x)\mathcal{P}(x, x')(r(x) + \gamma f_\Theta(x') - f_\Theta(x))\Phi(x).$$

The bias due to Markovian sampling is:

$$Z_t(\Theta) = \left(\hat{g}_t(\Theta) - \bar{g}(\Theta)\right)^\top(\Theta - \Theta^\star).$$

With the above definitions, the Lyapunov drift at time $t \geq 1$ can be bounded as follows:

$$\mathbb{E}_t^{++}\|\Theta(t+1) - \Theta^\star\|_2^2 \leq \|\Theta(t) - \Theta^\star\|_2^2 - 2\eta(1-\gamma)\|\mathcal{V} - f_{\Theta(t)}\|_\mu^2 + \eta^2\mathbb{E}_t^{++}[\|g_t(\Theta(t))\|_2^2\chi_t]$$
$$+ 2\eta\hat{g}^\top(\Theta(t) - \Theta^\star)\bar{\chi}_t + 2\eta Z_t(\Theta(t)),$$

where $\chi_t = 1 - \bar{\chi}_t = \mathbb{1}\{\|g_t(\Theta(t))\|_2 \leq b_t\}$. Compared to the case of iid sampling in Theorem 1, the difference is $Z_t(\Theta(t))$. In the following, we bound $\mathbb{E}Z_t(\Theta(t))$ by using the mixing time analysis in [4]. First, we provide two essential properties of $Z_t(\Theta)$ to verify the conditions in Lemma 10 in [4].

**Lemma 3.** *Under Assumption 1, we have:*

$$|Z_t(\Theta)| \leq (1 + 2\rho)^2, \ \Theta \in B_2(0, \rho), \tag{27}$$
$$|Z_t(\Theta) - Z_t(\Theta')| \leq 6(1 + 2\rho)^2\|\Theta - \Theta'\|_2^2, \ \Theta, \Theta' \in B_2(0, \rho). \tag{28}$$

Thus, we have:

$$\mathbb{E}Z_t(\Theta(t)) \le \mathbb{E}[Z_t(\Theta(t-\tau))] + 6(1+2\rho)^2 \mathbb{E}\|\Theta(t) - \Theta(t-\tau)\|_2. \tag{29}$$

We have the following inequality:

$$\|\Theta(t) - \Theta(t-\tau)\|_2 \le \sum_{k=t-\tau}^{t-1} \|\Theta(k+1) - \Theta(k)\|_2 \le \eta \sum_{k=t-\tau}^{t-1} \|g_t(\Theta(t))\|_2 \chi_t.$$

Taking the expectation above, and using Hölder's inequality:

$$\mathbb{E}\|\Theta(t) - \Theta(t-\tau)\|_2 \le \sum_{k=t-\tau}^{t-1} \left(\mathbb{E}[\|g_k(\Theta(k))\|_2^{1+p}]\right)^{\frac{1}{1+p}} \le \eta\tau u^{\frac{1}{1+p}}.$$

By using the information theoretic bound in Lemma 9 in [4], we obtain

$$\mathbb{E}Z_t(\Theta(t-\tau)) \le 2(1+2\rho)^2 \eta,$$

under the uniform ergodicity assumption in Assumption 1. Using the last two inequalities in (29), we obtain:

$$\mathbb{E}Z_t(\Theta(t)) \le 2(1+2\rho)^2 \left(1 + 6\tau u^{\frac{1}{1+p}}\right)\eta. \tag{30}$$

By using the above result, we obtain the ultimate inequality for the Lyapunov drift as follows:

$$\mathbb{E}\|\Theta(t+1) - \Theta^\star\|_2^2 \le \mathbb{E}\|\Theta(t) - \Theta^\star\|_2^2 - 2\eta(1-\gamma)\mathbb{E}\|\mathcal{V} - f_{\Theta(t)}\|_\mu^2 + \eta^2\mathbb{E}[\|g_t\|_2^2\chi_t] + 4\eta\rho\mathbb{E}[\|g_t\|_2\chi_t]$$
$$+ 4\eta^2(1+2\rho)^2 \left(1 + 6\tau u^{\frac{1}{1+p}}\right).$$

The proof follows from identical steps as Theorem 1. □

*Proof of Theorem 3.* The main idea in the proof is to establish a centering argument for both the bias (due to using clipped stochastic gradients) and the variability (controlled by $b_t$), and to use martingale concentration arguments based on Freedman's inequality and Azuma-Hoeffding inequality to bound the sample mean for the bias and variability, respectively. This strategy extends the approach in [6] for robust mean estimation to reinforcement learning, which has a dynamic behavior unlike the mean estimation problem. Namely, for any $t \in \{1, 2, \ldots, T\}$, we have:

$$\mathcal{L}(\Theta(t+1)) \le \mathcal{L}(\Theta(t)) - 2\eta(1-\gamma)\|f_{\Theta(t)} - \mathcal{V}\|_\mu^2$$
$$+ \eta^2\mathbb{E}[\|g_t(\Theta(t))\|_2^2 \mathbb{1}\{\|g_t(\Theta(t))\|_2 \le b_t\}|\mathcal{F}_t]$$
$$+ 2\eta B(t) + \eta^2 V(t),$$

where the first line follows from Lemma 2, and

$$B(t) = -\mathbb{E}[g_t(\Theta(t))^\top(\Theta(t) - \Theta^*)|\mathcal{F}_t] + g_t(\Theta(t))^\top(\Theta(t) - \Theta^*)\mathbb{1}\{\|g_t(\Theta(t))\|_2 \le b_t\}, \tag{31}$$

is the bias term, and

$$Z(t) = \|g_t(\Theta(t))\|_2^2 \mathbb{1}\{\|g_t(\Theta(t))\|_2 \le b_t\} - \mathbb{E}[\|g_t(\Theta(t))\|_2^2 \mathbb{1}\{\|g_t(\Theta(t))\|_2 \le b_t\}|\mathcal{F}_t]$$

is the variability. By telescoping sum over $t = 1, 2, \ldots, T$ and some algebraic manipulations, we have:

$$\frac{\mathcal{L}(\Theta(T+1))}{T} - \frac{\mathcal{L}(\Theta(1))}{T} \le -2\eta(\frac{1}{T}\sum_{t=1}^T f(\Theta(t)) - f(\Theta^*))$$
$$+ \eta^2 \frac{u}{T}\sum_{t=1}^T b_t^{1-p} + \frac{2\eta}{T}\sum_{t=1}^T B(t) + \frac{\eta^2}{T}\sum_{t=1}^T Z(t), \tag{32}$$

where we used (20) in the second line. In the following, we will bound the empirical processes $\frac{1}{T}\sum_{t=1} Z(t)$ and $\frac{1}{T}\sum_{t=1} B(t)$.

Note that $\{Z(t) : t \in \mathbb{N}\}$ is a martingale difference sequence (MDS) adapted to the filtration $\{\mathcal{F}_t : t \in \mathbb{N}\}$. Furthermore, note that

$$|Z(t)| \le 2b_t^2 \le 2b_T^2,$$

almost surely for any $t \leq T$. Thus, $\sum_{t=1}^{n} V(t)\mathbb{1}\{n \leq T\}$ forms a martingale with bounded differences, and by using Azuma-Hoeffding inequality [55], we have:

$$\frac{1}{T}\sum_{t=1}^{T} Z(t) \leq b_T\sqrt{\frac{L_\delta}{T}} = \frac{u^{\frac{1}{1+p}}T^{\frac{1-p}{2(1+p)}}}{L_\delta^{\frac{1-p}{2(1+p)}}} \leq \frac{u^{\frac{2}{1+p}}T^{\frac{1-p}{1+p}}}{L_\delta^{\frac{1-p}{1+p}}}, \tag{33}$$

with probability at least $1 - \delta/2$ where the last inequality holds since $T > L_\delta u^{-\frac{2}{1-p}}$.

We decompose $B(t)$ into predictable and non-predictable components as follows:

$$B(t) = \mathbb{E}[g_t(\Theta(t))^\top(\Theta(t) - \Theta^*)\mathbb{1}\{\|g_t(\Theta(t))\|_2 > b_t\}|\mathcal{F}_t] + B_0(t), \tag{34}$$

where the martingale difference sequence $B_0(t)$ is defined as follows:

$$B_0(t) = -\mathbb{E}[g_t(\Theta(t))^\top(\Theta(t) - \Theta^*)\mathbb{1}\{\|g_t(\Theta(t))\|_2 \leq b_t\}|\mathcal{F}_t]$$
$$+ g_t(\Theta(t))^\top(\Theta(t) - \Theta^*)\mathbb{1}\{\|g_t(\Theta(t))\|_2 \leq b_t\}. \tag{35}$$

By (25) (with $b_t$ replaced by $b_t$), the first term on the RHS of (34) is bounded by $2\rho u c_{t,\delta}^{-p}$. Thus, we have:

$$\frac{1}{T}\sum_{t=1}^{T} B(t) \leq \frac{2\rho u^{\frac{1}{1+p}}L_\delta^{\frac{p}{1+p}}}{T^{\frac{p}{1+p}}} + \frac{1}{T}\sum_{t=1}^{T} B_0(t). \tag{36}$$

In order to upper bound $\frac{1}{T}\sum_{t=1}^{T} B_0(t)$, we use Freedman's inequality for martingales [12]. To use Freedman's inequality, we verify the following conditions.

1. For any $t \leq T$, we have:

$$|g_t(\Theta(t))^\top(\Theta(t) - \Theta^*)\mathbb{1}\{\|g_t(\Theta(t))\|_2 \leq b_t\}| \leq 2\rho b_t \leq 2\rho b_t,$$

   almost surely.

2. The normalized quadratic variation process satisfies:

$$\frac{1}{T}\sum_{t=1}^{T} |B_0(t)|^2$$

$$\leq \frac{1}{T}\sum_{t=1}^{T} \mathbb{E}[|g_t(\Theta(t))^\top(\Theta(t) - \Theta^*)|^2\mathbb{1}\{\|g_t(\Theta(t))\|_2 \leq b_t\}|\mathcal{F}_t],$$

$$\leq \frac{4\rho^2}{T}\sum_{t=1}^{T} \mathbb{E}[\|g_t(\Theta(t))\|_2^2\mathbb{1}\{\|g_t(\Theta(t))\|_2 \leq b_t\}|\mathcal{F}_t],$$

$$\leq \frac{4\rho^2}{T}\sum_{t=1}^{T} u b_t^{1-p} \leq 4\rho^2 \frac{u^{\frac{2}{1+p}}T^{\frac{1-p}{1+p}}}{L_\delta^{\frac{1-p}{1+p}}},$$

   where the first inequality is due to $Var(Z) \leq \mathbb{E}[Z^2]$ for any random variable $Z$ with a finite variance, the second inequality follows from Cauchy-Schwarz inequality and triangle inequality with $\max\{\|\Theta(t)\|_2, \|\Theta^*\|_2\} \leq \rho$ due to projection, the third inequality follows from (20) with $b_t$ replaced by $b_t$.

Thus, by Freedman's inequality, we have:

$$\frac{1}{T}\sum_{t=1}^{T} B_0(t) \leq \frac{2\sqrt{2}\rho u^{\frac{1}{1+p}}L_\delta^{\frac{p}{1+p}}}{T^{\frac{p}{1+p}}} + \frac{4\rho L_\delta b_t}{3T},$$

$$\leq (2\sqrt{2} + 4/3)\rho\frac{u^{\frac{1}{1+p}}L_\delta^{\frac{p}{1+p}}}{T^{\frac{p}{1+p}}},$$

with probability at least $1 - \delta/2$. Therefore, from (36) and the above inequality, with probability at least $1 - \delta/2$, we have:

$$\frac{1}{T}\sum_{t=1}^{T} B(t) \leq \frac{7\rho u^{\frac{1}{1+p}} L_\delta^{\frac{p}{1+p}}}{T^{\frac{p}{1+p}}} \tag{37}$$

Hence, by substituting (33) and (37) into (32) with union bound, and using the specified step-size together with the facts that $\mathcal{L}(\Theta(1)) \leq 4\rho^2$ and $\mathcal{L}(\Theta(T+1)) \geq 0$, we conclude the proof. $\qquad\square$

## A.1 Finite-Time Analysis of Robust TD Learning without Realizability

In the following, we release the realizability assumption, and show that an additional function approximation error appears in the bounds for the general case.

**Theorem 5** (Performance of Robust TD learning – without realizability). Let

$$\epsilon_{\mathsf{app}} = \inf_{\Theta \in B_2(0,\rho)} \sqrt{\mathbb{E}|\mathcal{V}(x) - \langle \Theta, \Phi(x)\rangle|^2}.$$

Then, under Assumptions 1 and 2a we have the following bounds for Robust TD learning:

For $b_t = (ut)^{\frac{1}{1+p}}$ for any $t \in \mathbb{Z}_+$ and $\eta_t = \eta = \frac{2\rho(1-\gamma)}{(uT)^{\frac{1}{1+p}}}$, we have:

$$\mathbb{E}_{\substack{\Theta(1),\Theta(2),\dots,\Theta(T) \\ x\sim\mu}}\left[\left(\mathcal{V}(x) - \langle\bar{\Theta}(T), \Phi(x)\rangle\right)^2\right] \leq \frac{6\rho u^{\frac{1}{1+p}}}{(1-\gamma)T^{\frac{p}{1+p}}} + (\rho + \frac{1}{1-\gamma})\frac{2\epsilon_{\mathsf{app}}}{1-\gamma}, \tag{38}$$

for any $T > 1$.

*Proof of Theorem 5.* Note that we define

$$\Theta^\star \in \arg\min_{\Theta \in B_2(0,\rho)} \mathbb{E}[|\mathcal{V}(x) - \langle\Theta, \Phi(x)\rangle|^2].$$

By (19), we have:

$$\mathbb{E}[\mathcal{L}(\Theta(t+1))|\mathcal{F}_t] \leq \mathcal{L}(\Theta(t)) + 2\eta\mathbb{E}_t[g_t^\top(\Theta(t))(\Theta(t) - \Theta^\star)]$$
$$- 2\eta\mathbb{E}[g_t(\Theta(t))^\top(\Theta(t) - \Theta^*)\mathbb{1}\{\|g_t(\Theta(t))\|_2 > b_t\}|\mathcal{F}_t] + \eta^2 u b_t^{1-p}, \tag{39}$$

To eliminate the realizability assumption (Assumption 3), we make the following decomposition:

$$\mathbb{E}_t[g_t^\top(\Theta(t))(\Theta(t) - \Theta^\star)] = \mathbb{E}_t[(R_t + \gamma f_{\Theta(t)}(X_t') - f_{\Theta(t)}(X_t))(f_{\Theta(t)}(X_t) - \langle\Theta^\star, \Phi(X_t)\rangle)],$$
$$= \mathbb{E}_t\left[\left((\mathcal{T}f_{\Theta(t)})(X_t) - f_{\Theta(t)}(X_t)\right)\left(f_{\Theta(t)}(X_t) - \langle\Theta^\star, \Phi(X_t)\rangle\right)\right],$$
$$= \mathbb{E}_t\left[\left((\mathcal{T}f_{\Theta(t)})(X_t) - f_{\Theta(t)}(X_t)\right)\left(f_{\Theta(t)}(X_t) - \mathcal{V}(X_t)\right)\right]$$
$$+ \mathbb{E}_t\left[\left((\mathcal{T}f_{\Theta(t)})(X_t) - f_{\Theta(t)}(X_t)\right)\left(\mathcal{V}(X_t) - \langle\Theta^\star, \Phi(X_t)\rangle\right)\right].$$

Note that the last term corresponds to the approximation error, and we can bound its expectation as:

$$\mathbb{E}\left[\left((\mathcal{T}f_{\Theta(t)})(X_t) - f_{\Theta(t)}(X_t)\right)\left(\mathcal{V}(X_t) - \langle\Theta^\star, \Phi(X_t)\rangle\right)\right] \leq (1 - \gamma + 1)\|\mathcal{V} - f_{\Theta(t)}\|_\mu \cdot \epsilon_{\mathsf{app}}.$$

Since $|\mathcal{V}(x)| \leq \frac{1}{1-\gamma}$ and $|f_{\Theta(t)}(x)| \leq \rho$ for all $x$, and $\gamma \in (0,1)$, we have:

$$\mathbb{E}\left[\left((\mathcal{T}f_{\Theta(t)})(X_t) - f_{\Theta(t)}(X_t)\right)\left(\mathcal{V}(X_t) - \langle\Theta^\star, \Phi(X_t)\rangle\right)\right] \leq 2(\rho + \frac{1}{1-\gamma})\epsilon_{\mathsf{app}},$$

for any $t$. Using this result, and following identical steps as Theorem 1 with the same step-size yields the result. $\qquad\square$

# B   Proofs for Robust Natural Actor-Critic

*Proof of Theorem 4.* We use the following Lyapunov function for the analysis [1, 57, 34]:

$$\mathcal{L}(\pi) = \sum_{s \in \mathbb{S}} \mathsf{d}_\lambda^{\pi^\star}(s) \sum_{a \in \mathbb{A}} \pi^\star(a|s) \log \frac{\pi^\star(a|s)}{\pi(a|s)}. \tag{40}$$

For the Lyapunov drift, at any iteration $k$, we have:

$$\mathcal{L}(\pi_{k+1}) - \mathcal{L}(\pi_k) = \sum_{s,a} \mathsf{d}_\lambda^{\pi^\star}(s) \pi^\star(a|s) \log \frac{\pi_k(a|s)}{\pi_{k+1}(a|s)}. \tag{41}$$

Since $\sup_{s,a} \|\Phi(s,a)\|_2 \leq 1$, $\nabla_W \log \pi_W(a|s)$ is 1-Lipschitz continuous [1]. Thus, we have:

$$|\log \pi_{k+1}(a|s) - \log \pi_k(a|s) - \nabla^\top \log \pi_k(a|s)(W(k+1) - W(k))| \leq \frac{1}{2} \|W(k+1) - W(k)\|_2^2. \tag{42}$$

Since $W(k+1) = W(k) + \alpha \bar{\Theta}_k(T)$, we have:

$$\mathcal{L}(\pi_{k+1}) - \mathcal{L}(\pi_k) \leq \frac{\eta^2}{2} \|\bar{\Theta}_k(T)\|_2^2 - \eta \cdot \mathsf{d}_\lambda^{\pi^\star}(s) \pi^\star(a|s) \nabla^\top \log \pi_k(a|s) \bar{\Theta}_k(T). \tag{43}$$

By performance difference lemma [26], we have:

$$\mathcal{V}^\pi(s) - \mathcal{V}^{\pi'}(s) = \frac{1}{1-\gamma} \mathop{\mathbb{E}}_{\substack{s \sim \mathsf{d}_\lambda^\pi \\ a \sim \pi(\cdot|s)}} [\mathcal{A}^{\pi'}(s,a)]. \tag{44}$$

Using the last two inequalities, we have the drift inequality:

$$\mathcal{L}(\pi_{k+1}) - \mathcal{L}(\pi_k) \leq \frac{\eta^2}{2} \|\bar{\Theta}_k(T)\|_2^2 - \eta \sum_{s,a} \mathsf{d}_\lambda^{\pi^\star}(s) \pi^\star(a|s) \Big( \nabla^\top \log \pi_k(a|s) \bar{\Theta}_k(T) - \mathcal{A}^{\pi_k}(s,a) \Big)$$
$$- \eta \Big( \mathcal{V}^{\pi^\star}(\lambda) - \mathcal{V}^{\pi_k}(\lambda) \Big).$$

For the log-linear policy parameterization, we have

$$\nabla \log \pi_W(a|s) = \Phi(s,a) - \sum_{a' \in \mathbb{A}} \pi_W(a'|s) \Phi(s,a').$$

Also, from the definition of $\mathcal{A}^\pi(s,a) = \mathcal{Q}^\pi(s,a) - \sum_{a' \in \mathbb{A}} \pi(a'|s) \mathcal{Q}^\pi(s,a')$,

$$\mathbb{E}\Big[\Big(\nabla^\top \log \pi_k(a|s) \bar{\Theta}_k(T) - \mathcal{A}^{\pi_k}(s,a)\Big)^2\Big] \leq \mathbb{E}\Big[\Big(\langle \Phi(s,a), \bar{\Theta}_k(T) \rangle - \mathcal{Q}^{\pi_k}(s,a)\Big)^2\Big],$$
$$= \sum_{s,a} \mathsf{d}_\lambda^{\pi^\star}(s) \pi^\star(a|s) \Big(\langle \Phi(s,a), \bar{\Theta}_k(T) \rangle - \mathcal{Q}^{\pi_k}(s,a)\Big)^2,$$
$$\leq C_{\text{conc}} \sum_{s,a} \mu^{\pi_k}(s,a) \Big(\langle \Phi(s,a), \bar{\Theta}_k(T) \rangle - \mathcal{Q}^{\pi_k}(s,a)\Big)^2,$$

where the first line follows from the fact that $Var(X) \leq \mathbb{E}[X^2]$ for any random variable $X$ with finite second-order moments, and the last line follows from a change of measure argument. Then, by Theorem 3, we have:

$$\sum_{s,a} \mu^{\pi_k}(s,a) \Big(\langle \Phi(s,a), \bar{\Theta}_k(T) \rangle - \mathcal{Q}^{\pi_k}(s,a)\Big)^2 \leq \frac{\rho u_k^{\frac{1}{1+p}}}{(1-\gamma)T^{\frac{p}{1+p}}} \Big(3L_\delta^{-\frac{1-p}{2(1+p)}} + 7L_\delta^{\frac{p}{1+p}}\Big),$$

with probability at least $1 - \delta/K$. Furthermore, we have:

$$\|\bar{\Theta}_k(T)\|_2^2 \leq \rho^2,$$

for any $k, T$ by the projection. As such, we can bound the drift inequality as follows:

$$\mathcal{L}(\pi_{k+1}) - \mathcal{L}(\pi_k) \leq \frac{\eta^2}{2} \rho^2 + \eta \sqrt{C_{\text{conc}} \frac{\rho u_k^{\frac{1}{1+p}}}{(1-\gamma)T^{\frac{p}{1+p}}} \Big(3L_\delta^{-\frac{1-p}{2(1+p)}} + 7L_\delta^{\frac{p}{1+p}}\Big)}$$
$$- \eta(1-\gamma)\Big(\mathcal{V}^{\pi^\star}(\lambda) - \mathcal{V}^{\pi_k}(\lambda)\Big), \tag{45}$$

with probability at least $1 - \delta/K$. By telescoping sum of the above inequality, using union bound, and noting that $\pi_0(a|s) = \frac{1}{|\mathbb{A}|}$ for any $s, a$, which leads to $\mathcal{L}(\pi_1) = \log|\mathbb{A}|$, we conclude the proof. $\qquad\square$

