# OpenReview forum: "Provably Robust Temporal Difference Learning for Heavy-Tailed Rewards"
_NeurIPS.cc/2023/Conference — NeurIPS 2023 poster_

### Official Review · Reviewer_GiX9 · 2023-07-06

**Soundness:** 2 fair
**Presentation:** 2 fair
**Contribution:** 2 fair
**Rating:** 4
**Confidence:** 2

**Summary:**

This paper studies the problem of reinforcement learning under heavy-tailed reward. The paper present two algorithms: robust TD learning and robust NPG that achieves polynomial sample complexity in the generative setting.

**Strengths:**

The problem of heavy tail reward is important and understudied in RL.

**Weaknesses:**

While the paper is well-written, I'm having trouble appreciating the technique contribution in this paper. Could the authors elaborate what techniques/results in the paper goes beyond a straight-forward combination of (1) robust mean estimation under heavy tail distribution and (2) existing analysis of TD learning and NPG?

**Questions:**

See above.

---

> ### Author Rebuttal · Authors · 2023-08-10
>
> We thank the reviewer for the feedback and questions. Please refer to our general response above for our contributions, and differences of our work with the existing papers.

---

> > ### Comment · Reviewer_GiX9 · 2023-08-10
> > **Thank you for the elaboration of contribution**
> >
> > Thank you to the authors for elaborating the contribution in the paper. From my experience, extending existing robust mean estimation results to Martingale sequences are mostly trivial, because most concentration inequalities for i.i.d. data have equivalent counterpart for martingale sequence, including the ones used in this paper. Adjusting the clipping radius for clipped mean method is also a standard component in the clipped mean analysis, even in the standard i.i.d. setting. Therefore, I still believe that the technique contribution in this paper is limited.
> >
> > Despite that, tools for TD learning with heavy-tailed rewards is a valuable addition to the RL toolkit, so I'm not in strong favor of accepting or rejecting this manuscript.

---

> > > ### Author Response · Authors · 2023-08-15
> > >
> > > We thank the reviewer for the response.
> > >
> > > Without doubt, the perceived degree of difficulty in any piece of research is a subjective judgement, depending on the particular background and expertise of each reader. As such, we respect the reviewer’s evaluation of our contribution. However, probably due to the reviewer's specific expertise, we believe that the reviewer’s “trivial” characterization of the technical component may be over-simplifying the challenges involved with carefully completing the steps necessary to prove the results in our work. Such an evaluation has the danger of dismissing a great amount of valuable works in the literature with similar claims.
> > >
> > > To the best of our knowledge, this is the first work that successfully manages the heavy-tailed impact in the TD learning and actor-critic domain with a precise analysis of its guarantees. We note that we analyze a complicated stochastic iterative algorithm, which is considerably more complicated than mean estimation. As such, we believe that it makes a good contribution to the RL domain and is worthy of publication in this conference. We leave to all the reviewers and the chairs to make the final judgement on that. Thank you.

---

### Official Review · Reviewer_jt3m · 2023-07-06

**Soundness:** 3 good
**Presentation:** 3 good
**Contribution:** 3 good
**Rating:** 6
**Confidence:** 3

**Summary:**

This paper studies the TD and NAC with heavy tailed reward noise. The authors provide both theoretical results and the numerical verification to support their claims.

**Strengths:**

- This paper is well written and easy to follow. The numerical results firmly support their theoretical claims and show the advantage of the robust algorithm. The theoretical results seem correct to me.
- The authors suggest several practical cases where heavy-tailed reward noises are crucial and the robustness estimation algorithm can help to improve the performance.

**Weaknesses:**

- Comparing with the previous TD learning results [4], it's not clear to me what's the major contribution of this paper. Is it simply combining the result and analysis in [4] and the concentration analysis for heavy-tailed noise (usually the robust-mean results)?
- The algorithm parameter needs the knowledge of $p$, where I wonder how to estimate it in practice.

**Questions:**

- My major concern on this paper is in the weakness section regarding the contribution. It could be better highlighted if the authors can demonstrate some non-trivial processes obtaining this $p / (1 + p) $ style result for the heavy-tailed noise.

---

> ### Author Rebuttal · Authors · 2023-08-10
>
> We thank the reviewer for the feedback and questions. Below, we provide our answers to the comments in the Weaknesses section.
>
> - *W1: Contributions.* Please refer to our general response for our contributions and the differences between our paper and existing works in the literature.
>
> - *W2: Knowledge of $p$.* In our work, we assumed that moments of order $1+p$ for some $p\in(0,1)$ exist. $p$ is estimated in practice by using the celebrated Hill estimator in extreme value theory (Resnick and Starica, 1998; Gomes et al., 2000). We also note that the exact knowledge of the tail index is not necessary. For example, for $R \sim Pareto(1, 1+p_0)$ distributed rewards for some $p_0\in(0,1]$, an upper bound $u$ on $\mathbb{E}[|R|^{1+p}]$ can be used in the algorithm for any $p\in(0,p_0)$. The performance of the algorithm depends on the accuracy of $u$ as an upper bound to the raw moments of order $1+p$.
>
> Our answer to the question raised by the reviewer is as follows.
>
> - *Question 1.* We thank the reviewer for pointing this out. A wide class of processes including Pareto distribution with parameter $\alpha\in(1,2)$ satisfy the moment assumptions in our paper, among other power-law distributions with finite moments of order $\alpha\in(1,2]$. Please also refer to our general response. We will clarify this part in the updated version.
>
> **References:**
>
> S. Resnick and C. Stărică, "Tail index estimation for dependent data." The Annals of Applied Probability 8.4: 1156-1183 (1998)
>
> C. P. Gomes et al., "Heavy-tailed phenomena in satisfiability and constraint satisfaction problems." Journal of automated reasoning 24.1-2: 67-100 (2000).

---

> > ### Comment · Reviewer_jt3m · 2023-08-19
> >
> > Thank the authors for their response. I do not have any further concerns now. I decided to keep my score

---

> > > ### Author Response · Authors · 2023-08-21
> > >
> > > We would like to thank the reviewer for the feedback and questions.

---

### Official Review · Reviewer_db5F · 2023-07-14

**Soundness:** 3 good
**Presentation:** 3 good
**Contribution:** 3 good
**Rating:** 7
**Confidence:** 2

**Summary:**

This work discusses the failure of classical TD learning under heavy-tailed reward. This phenomenon motivates the author to propose a new technique, robust TD learning, with a dynamic gradient clipping mechanism to overcome this issue. The theoretical convergence rates for both in-expectation and high-probability cases are provided to ensure the convergence of this new algorithm under heavy-tailed reward. Some empirical experiments are taken to verify the theoretical result.

**Strengths:**

The question raised at the beginning of this paper, "Can temporal difference learning with function approximation be modified to provably achieve global optimality under stochastic rewards with heavy tails?" is well addressed by proposing a new algorithm and providing the convergence upper bounds. According to the related works, this problem has not been solved before.

Regarding the quality and clarity, the whole paper is complete and well-presented except for some minor issues.

The motivation example is clear, and it well justifies the issue that this paper aims to solve, which makes this paper well-motivated on the theoretical side. Also, two applications are provided and formulated in the described senario. I believe this paper has made a clear contribution.



**Weaknesses:**

Lack of discussions on the assumptions: I agree that these assumptions are very standard in the convergence analysis of TD-based algorithms, but it is necessary to add more backgrounds. More explicitly, about the definition of heavy-tailed, some well-known heavy-tailed distributions may have infinite expectation (e.g. Cauchy distribution and Pareto distribution Pareto(1, 1)). This make me confused about Assumption 1.2.

Presentation: The parameter $p$ has never been clearly explained until Assumption 1 (on page 4), though it has been used in Main contributions and Abstract.




**Questions:**

1. It seems that some heavy-tailed distributions cannot satisfy both Assumption 1.2 and Assumption 1.3. For example, Cauthy distribution and Pareto(1, 1) distribution, because they have an infinite expectation. Do I misunderstand something here?

2. Does Theorem 3 directly imply Theorem 2? Why is it necessary to derive two bounds here? If I simply take an expectation on both sides of Theorem 3: (1) with the probability $1-\delta$, it has the same rate as Theorem 2; (2) with the probability $delta$, bounding everything by a sufficiently large constant. Then tuning the parameter $delta$ could lead to the in-expectation bound.

**Limitations:**

This is a purely theoretical paper so no limitations and potential negative societal impact.

---

> ### Author Rebuttal · Authors · 2023-08-10
>
> We thank the reviewer for the feedback and questions. Below, we provide our answers.
>
> - *Question 1.* Regarding Assumption 1.2, please also see our general response above. As the reviewer pointed out, our Assumption 1.2 implies that the random reward has finite mean and potentially infinite variance. Many important probability distributions that are used for statistical modeling in computing and communication systems satisfy this assumption. For example, Pareto distributions $\mathbb{P}(X > x)\leq c/x^{1+p}$ for some $p\in(0,1)$ satisfy the aforementioned assumption, among other power-law distributions with finite mean and potentially infinite variance.
>
> - *Question 2.* We thank the reviewer for this question. The main difference between Theorem 2 and Theorem 3 is about sampling. In Theorem 3, we proved the result under iid sampling (Assumption 2a), while Theorem 2 is under more general Markovian sampling (Assumption 2b).

---

> > ### Comment · Reviewer_db5F · 2023-08-17
> >
> > Thanks for the answer! My concerns are solved and I have no further questions for now.

---

> > > ### Author Response · Authors · 2023-08-17
> > >
> > > We would like to thank the reviewer very much for the feedback and comments, and we are very glad that all concerns are solved.

---

### Official Review · Reviewer_z9uh · 2023-07-26

**Soundness:** 3 good
**Presentation:** 2 fair
**Contribution:** 2 fair
**Rating:** 5
**Confidence:** 2

**Summary:**

The paper studies the value function estimation problem when the reward of the MDP is heavy tail. A robust version of TD learning is proposed which introduces a clipping step to the algorithm. The authors give both the theoretical analysis and numerical experiments for the proposed algorithm. And they further applied the robust TD learning algorithm to the policy optimization problem, and give a theoretical analysis for it.

**Strengths:**

The paper studies an important and interesting problem. The presentation is clear and the arguments are well supported by both theoretical and empirical evidence.

**Weaknesses:**

The presentation can be improved. In algorithm 1, the variables have a subscript k which is not needed in this algorithm. The proof part skips some details, such as the end of the proof of Theorem 1. Some notations are not defined, such as the total variation and the \mu-norm. The analysis shares a similar framework with that in [4], which limits the theoretical novelty of the analysis. The result mainly applies to the i.i.d. sampling case. And due to the mixing time assumption, the Markovian sampling case is essentially the same as the i.i.d. case, as one can easily get a near-independent sequence by subsampling. The experiments are conducted on artificial problems, and real-world application related experiments are not presented in the paper. The experiments are only about the value estimation, but do not solve the downstream policy optimization task.

**Questions:**

In the experiment results, the curve for TD learning has a rapid increase in error at the initial phase. This is an interesting phenomenon given that the curve represents the averaged performance. Could the authors provide any comments on it?

**Limitations:**

There is not a section about limitation. Since the work is mainly focused on the theoretical analysis, it shouldn't have a direct negative social impact.

---

> ### Author Rebuttal · Authors · 2023-08-10
>
> We thank the reviewer for the comments and questions.
>
> - We refer to our general response above for our contributions, and for differences with [4] (i.e., Bhandari et al., 2018).
>
> - We thank the reviewer for pointing out the typo, we will remove the subscript $k$ in the updated version.
>
> Below, we provide our answer to the question raised by the reviewer.
>
> - *Question 1.* In the heavy-tailed case, there is a large probability that the gradient noise will be very large. Initially, since the number of samples (i.e., number of TD learning iterations) is small, this very large gradient noise will cause a significant deviation in different directions than the descent direction even for the average-iterate, causing a large mean-squared error initially. As the number of iterations increase, the gradient noise will be canceled with the help of the gradient clipping and point the direction of improvement, which leads to vanishing error eventually.

---

> > ### Comment · Reviewer_z9uh · 2023-08-14
> >
> > Thank you for your response. My score will remain the same.

---

> > > ### Author Response · Authors · 2023-08-15
> > >
> > > We would like to thank the reviewer for the feedback and comments.

---

### Official Review · Reviewer_ZkyR · 2023-07-26

**Soundness:** 3 good
**Presentation:** 4 excellent
**Contribution:** 2 fair
**Rating:** 7
**Confidence:** 3

**Summary:**

The authors propose and study variants of TD(0) and Natural Actor-Critic that provably converge under (on-policy in the case of TD(0)) linear function approximation when the rewards may not have finite second moment or variance, but instead a finite (1+p) moment for $p \in (0,1]$; the authors use this as the definition of "heavy-tailed" rewards.

The techniques used include:
- Clipping the update direction using a decaying threshold
- Using Holder's inequality + the decaying threshold to control the bias introduced by clipping the update direction, where in the normal analysis, one would upper-bound the expected noise by zero.

The rest of the techniques are otherwise similar to well-known previous work, in particular by Bhandari et al. 2018 (at least in essence, e.g., there is no difference in doing a Lyapunov analysis or using the Online-Gradient-Descent update bound for $g_t$ and $\Theta_t$). Additional assumptions used include:
1. realizability, and
2. knowledge of a projection radius containing the optimum.

The authors complement the analysis with numerical results.

**Strengths:**

The paper is very well-written. The logic follows clearly and easily. Examples are given when appropriate. The assumptions are laid out clearly. It's a nice, easy-to-follow read for a theoretician.

The paper contains the typical interesting (even if standard) results: the robust step-size case, the $1/t$ step-size case with knowledge of $\lambda_\textnormal{min}$, the simulator/i.i.d. sample case, and the Markov sample case under uniform exponential mixing.

When $p=1$, the results naturally fall back to the $O(1/\sqrt{T})$ and $O(1/T)$ rates of [4].

Considering NAC strengthens the paper, covering both policy-evaluation and control.

**Weaknesses:**

From a technical point of view, the assumptions are rather strong. Specifically,

- Realizability is assumed, which is not really required for this MRP setting.
- In addition, it is assumed that we have knowledge of an upper-bound $\rho$ on the norm of the optimal $\Theta^*$. This is not required by e.g., Bhandari et. al. 2018.
- "Heavy-tail" is defined as satisfying a relaxed version of the usual (non-heavy-tailed) assumption: degree 2 is replaced by degree $1+p$ for $p \in (0,1]$. In addition, it is assumed that this value $p$ is known. Does this definition suite the examples of heavy-tailed rewards given in the introduction?

In terms of related work, I am not entirely convinced that (as written on line 112), the cases of SGD and RL really differ here, at least as far as the main technique for introducing and handling bias (diminishing threshold + Holders on top of the noise term) is concerned.

Finally, a recent (perhaps concurrent?) work (https://arxiv.org/pdf/2205.08464.pdf), considers defining and using robust losses instead of gradient-clipping. The merits and shortcomings of the two different approaches are worth discussing. The references therein also point to previous work (without theoretical guarantees) that use clipping of the rewards, the errors, or the update directions.

**Questions:**

Please comment on the weaknesses mentioned above. In particular:

- The strength of the heavy-tailed assumption.
- How hard it is to relax the projection onto ball of radius $\rho$ for each of the iid and Markov cases?
- Do you  need the realizability assumption if you instead bound the loss defined by $\phi^T \theta*$?

**Limitations:**

The technical limitations were already discussed in the "Weaknesses" and "Questions" section. This is a theoretical work so wider societal impacts are not apparent and addressable at this level.

---

> ### Author Rebuttal · Authors · 2023-08-10
>
> We would like to thank the reviewer for the comments and questions. Below, we provide answers to the questions.
>
> - *Question 1.* We have provided our comments regarding the realizability assumption in our general response above.
>
> - *Question 2.* We thank the reviewer for this important question. While projection onto a general set can be computationally complex, fortunately projection onto a simple structure such as a $\ell_2$-ball around 0 with radius $\rho$ is a low-complexity operation. As such, this is considered in our design. Alternatively, the projection step can be eliminated by using early stopping methods as in (Cayci et al., 2023). We can add a comment related to this in our update.
>
> - *Question 3.* Exactly. If the error is defined with respect to the best in-class approximator $\langle\Phi(x), \Theta^\star\rangle$ rather than the true value function, the realizability assumption can be eliminated.
>
> We provide our responses to the comments and questions raised by the reviewer in the Weaknesses section below. We would like to thank the reviewer for the reference. We will cite the reference in the updated version with appropriate commentary.
>
> - Realizability can be assumed as we discussed in our general response, by incorporating the approximation error into the error bound
> in the general case.
>
> - As the reviewer pointed out, the techniques outlined in Section 8.2 of (Bhandari et al., 2018) regarding the choice of projection radius $\rho$ can be used in our work as well. We will comment on this in our update.
>
> - Our statistical assumption on heavy-tailed rewards, i.e., existence of moments of order $1+p$ for some $p\in(0,1]$, is satisfied by heavy-tailed distributions that are traditionally used for modeling, such as Pareto distribution with parameter $\alpha\in(1,2]$. Such power-law distributions are widely used in modeling various phenomena, including job sizes and delay statistics in computing and communication systems (Harchol-Balter, 2002) (Wei and He, 2007). We have also provided a comment on this in our general response.
>
> **References.**
>
> M. Harchol-Balter, "Task Assignment with Unknown Duration." Journal of the ACM 49.2: 260-288 (2002).
>
> Z. Wei and J. He, "Modeling end-to-end delay using pareto distribution." Second International Conference on Internet Monitoring and Protection (ICIMP 2007). IEEE, 2007.
>
> S. Cayci et al., "Sample complexity and overparameterization bounds for temporal difference learning with neural network approximation." IEEE Transactions on Automatic Control, vol. 68, no. 5, pp. 2891-2905 (2023).

---

> > ### Comment · Reviewer_ZkyR · 2023-08-20
> >
> > Thanks for your response, which has addressed some of my concerns. I still think this is a well-written paper.
> >
> > I think the answer to Question 2 does not completely address what I was asking. My question is not about "relaxing" in the sense of making it computationally efficient. What I mean is: can you remove the projection without introducing other techniques like early stopping? The work of Bhandari 2018 does this. I wonder why you need this radius in the analysis, given that you are already clipping the gradient, hence its norm is controlled (context: the radius is usually used for controlling the norm of the gradient / semi-gradient). Is this true that you only need this in the Markov noise case for handling the non-zero-mean error there, or would you also need the projection if the transitions are drawn i.i.d.?
> >
> > As for the heavy-tailed definition, my question was: do we know $p$ easily for most applications of interest? I see that you have answered this in response to Reviewer jt3m. Thank you!

---

> > > ### Comment · Reviewer_ZkyR · 2023-08-20
> > >
> > > Also, I appreciate your answer regarding the realizability assumption. Thanks for the clarification.
> > > (
> > > Please note that in the RL theory literature, realizability is assumed mainly when one wants to discuss inherent complexity of the problem and sample-complexity and in lower-bound results. For this reason, making this assumption in your paper - which is already quite clearly written - for the sake of simplifying the presentation actually contributes to a confusion of realizability being "necessary" in a restrictive sense.
> > > )

---

> > > ### Author Response · Authors · 2023-08-21
> > >
> > > We would like to thank the reviewer for the response.
> > >
> > > - For Question 2, we would like to thank the reviewer for the clarification. As the reviewer stated, our work employs an explicit regularization method (e.g., $\ell_2$-projection, early stopping, etc.) to control $\|\Theta(t)\|_2$. We used explicit regularization mainly to control the bias introduced by dynamic gradient clipping to control heavy-tailed semi-gradients (we refer to Equation (24)) under both iid and Markovian sampling. Our use of explicit regularization to handle gradient bias and heavy-tailed semi-gradients constitutes the main difference with (Bhandari et al., 2018), which employs explicit regularization only to handle Markovian sampling in the light-tailed case with bounded semi-gradients. We will add a discussion on this in the updated version for clarification. (As a final note, the use of dynamic gradient clipping without projection also provides an upper bound $\|\Theta(t)\|$ by triangle inequality, but this bound is too crude and useless to establish near-optimality, i.e., grows at a very large rate over $t$ since we need to increase the clipping radius $b_t$ dynamically over $t$ to eliminate the bias introduced by the gradient clipping. As such, explicit regularization yields a more direct and effective control over $\|\Theta(t)\|_2$.)
> > >
> > > - For the question regarding the knowledge of $p$, we would like to thank the reviewer very much, and we are glad that the concern is solved.
> > >
> > > - Regarding the realizability assumption, we are very glad that the concern is now solved. We would like to thank the reviewer very much for the suggestion. To address the reviewer's suggestion, we will expand our discussion, and provide results without the realizability assumption (i.e., by including the approximation error) in our update.

---

### Official Review · Reviewer_pbb6 · 2023-07-27

**Soundness:** 3 good
**Presentation:** 3 good
**Contribution:** 2 fair
**Rating:** 5
**Confidence:** 3

**Summary:**

This paper discusses robust Temporal Difference (TD) methods for RL with heavy-tailed rewards. Specifically, it proposes robust variants of TD(0) and Natural Actor-Critic (NAC) for policy evaluation and RL control, respectively, and then derives their average iterate's convergence rates.

**Strengths:**

1. Convergence rates for TD(0) and NAC with function approximation are obtained in the non-trivial heavy-tailed reward setup.

2. The above results have been obtained both in the generative as well as the Markovian noise model.

**Weaknesses:**

1. Assumption 3 is extremely strong. While the authors mention that there would be an additional non-vanishing approximation error when this assumption is not the case, it is unclear what the point of convergence would be in that case.

2. No reasons/insights have been provided on why clipping helps deal with the heavy-tailed nature.

3. In the related works section, it is mentioned that existing theoretical analyses for TD learning only provide guarantees in the expected error rather than in high probability. This is incorrect; e.g., see

a.) Patil, G., Prashanth, L.A., Nagaraj, D. and Precup, D., 2023, April. Finite time analysis of temporal difference learning with linear function approximation: Tail averaging and regularisation. In International Conference on Artificial Intelligence and Statistics (pp. 5438-5448). PMLR.

b.) Dalal, G., Szörényi, B., Thoppe, G. and Mannor, S., 2018, April. Finite sample analyses for TD (0) with function approximation. In Proceedings of the AAAI Conference on Artificial Intelligence (Vol. 32, No. 1).

**Questions:**

1. Why should robust NAC converge to the optimal policy (in the value function sense)? I am not well-versed in the actor-critic literature, but is global convergence straightforward even in the function approximation setting? In the case of SARSA with linear-function approximation (Melo-Meyn-Ribeiro, 2008), one required that the policy improvement operator be Lipshitz with a `small' Lipschitz constant to ensure a unique fixed point of the Bellman operator. For example,  this holds for soft-max type policies with a sufficiently small inverse temperature parameter. In this work though, no such assumption is made apart from full realizability of the function approximation space.

Melo, F.S., Meyn, S.P. and Ribeiro, M.I., 2008, July. An analysis of reinforcement learning with function approximation. In Proceedings of the 25th international conference on Machine learning (pp. 664-671).

2. Is the obtained convergence rate of $\tilde{O}(T^{-p})$ tight for $p < 1?$

Minor:
3. What is k in the subscript of $\theta$ in Algorithm 1?

4. In line 199, should $\bar{\chi}(t)$ in the definition of $Z(t)$ be $\chi(t)?$

4. In the displayed equation above line 482, what is $V(t)?$

**Limitations:**

I could not find any limitations section.

---

> ### Author Rebuttal · Authors · 2023-08-10
>
> We would like to thank the reviewer for the questions and comments. Below, we provide our responses to the reviewer's comments and questions in *Weaknesses* and *Questions*, respectively.
>
> - *W1: Assumption 3.* We thank the reviewer for pointing this out. Please note that we have also addressed this question in our general response above. The goal in TD learning with function approximation is to minimize the error $\mathcal{L}(\Theta)$ over the parameter $\Theta$. If the value function $\mathcal{V}$ does not lie in the class of functions of type $f_\Theta$, there will be a non-vanishing function approximation error $\mathcal{L}(\Theta^\star)$ where $\Theta^\star$ is the optimal parameter, we characterized this approximation error in the general response. This approximation error is unavoidable, and appears in the seminal works (Tsitsiklis and van Roy, 1997) and (Bhandari et al., 2018) as well, as these works consider convergence to the fixed point of the *projected* Bellman equation rather than the true value function, which is the fixed point of the Bellman equation. Our convergence results in this paper shows that all statistical errors vanish with $T$, and the optimal total error $\mathcal{L}(\Theta^\star)$ is obtained as $T\rightarrow\infty$ at the rates we specified in the submission under a given linear function approximation architecture.
>
> - *W2: Intuitive explanation on how gradient clipping tames heavy tails.* As the reviewer suggested, we will expand the discussion in Line 210 on how gradient clipping helps taming heavy-tailed gradient noise in the updated version. We note that clipping bounds the gradient, thus making its variance bounded, but this comes at the expense of introducing a bias in the gradient update. By dynamically increasing the clipping radius over time carefully, the bias is gradually eliminated while controlling the variance, which establishes convergence.
>
> - *W3: Related works.* We would like to thank the reviewer for the references, we will cite these references in the updated version with appropriate commentary. In addition to noting the relationships, we will clarify that the focus in our work is the heavy-tailed regime, where we aim to establish high-probability bounds.
>
> In the following, we provide our responses to the questions raised by the reviewer.
>
> - *Question 1.* Policy gradient methods, which contain *natural actor-critic (NAC)* algorithm that we investigate in our work, aim to find the optimal policy within a restricted policy class directly by using gradient-based optimization methods. Depending on the richness of this policy class (e.g., rank of the feature matrix, ambient dimension in our case), the optimal policy is approximated by the output of the policy optimization algorithm. An assumption to establish convergence of the NAC method to global optimum is the concentrability coefficient assumption (Assumption 4 in our paper), which implies that the state-action pairs visited by the optimal policy should be covered by the support of the state-action visitation distributions under the policy iterates. This assumption is standard in the policy gradient literature (e.g., Agarwal et al., 2021; Wang et al, 2019; Yuan et al., 2022; Scherrer et al., 2015, among many others, and also see Xie et al., 2022 on its role). Sufficient exploration and an initial state distribution with sufficiently large support set implies this assumption in general.
>
> - *Question 2.* The tightness of the bounds requires lower bounds for the statistical errors in the heavy-tailed regime $p < 1$. It is definitely an important and complicated question that we leave as a future work.
>
> - *Questions 3-5.* We thank the reviewer for pointing out the typos. The subscript $k$ is unnecessary in Algorithm 1, $\bar{\chi}(t)$ should be replaced by $\chi(t)$ in Line 199, and $V(t)$ should be $Z(t)$ (the variability) in Line 482. We will correct these typos in the updated version.
>
> **References.**
>
> R. Yuan, et al., "Linear convergence of natural policy gradient methods with log-linear policies." arXiv preprint arXiv:2210.01400 (2022).
>
> L. Wang et al., "Neural policy gradient methods: global optimality and rates of convergence." International Conference on Learning Representations (2019).
>
> B. Scherrer et al., “Approximate modified policy iteration and its application to the game of tetris.” J. Mach. Learn. Res., vol. 16, pp. 1629–1676 (2015).
>
> A. Agarwal et al., "On the theory of policy gradient methods: Optimality, approximation, and distribution shift." The Journal of Machine Learning Research 22.1: 4431-4506 (2021).
>
> T. Xie, et al., "The role of coverage in online reinforcement learning." arXiv preprint arXiv:2210.04157 (2022).

---

> > ### Comment · Reviewer_pbb6 · 2023-08-13
> >
> > Thanks for your detailed response (in the rebuttal and the general post above). Nevertheless, it is still unclear what the point of convergence would be in the function approximation setting when the realizability assumption does not hold.  Correct me if I am wrong, but the definition of $\epsilon_{app}$ above depends on the choice of a particular starting state $x.$ What is this state $x?$ More broadly, how close and, also importantly under which norm, would your estimate be to the whole value-function vector $V?$ If your estimate is not close enough, it is unclear to me how useful the results of this work are.

---

> > > ### Author Response · Authors · 2023-08-15
> > >
> > > We thank the reviewer for this insightful question. We believe this is a very important question, aiming to get to the core of the contribution, even when the realizability assumption does not hold.
> > >
> > > The stationary state distribution of the Markov process is $\mu$, and we aim to learn the value function $\mathcal{V}(x)$ for every $x\in\mathbb{X}$ by using samples. The standard error metric used in TD learning analysis is $\mu$-weighted $\ell_2$-error, defined as $\sqrt{\sum_{x\in\mathbb{X}}\mu(x)\big(f(x)-\mathcal{V}(x)\big)^2}$, where the weight distribution $\mu$ is the stationary distribution of the Markov process, measuring the frequency of visits to each state in the long run. Intuitively, the error made in a frequently-visited state in the long run (as measured by $\mu$) is weighted more under this error metric. For the mathematical properties of this error and its relation to the Bellman operator, we refer to Chapter 6 in (Bertsekas, 2011) and (Tsitsiklis and van Roy, 1997). We note that visits to each state in the long-run is important, and a single trajectory with an arbitrary initial state from the Markov reward process can be used to learn the value function by using TD learning.
> > >
> > > Given these, without the realizability assumption, the TD learning error can be decomposed into **(i)** the statistical error that measures the error we make in learning the best function approximator for $\mathcal{V}$ from samples under the $\mu$-weighted $\ell_2$-norm within the class of linear functions with features vectors $\Phi$, and **(ii)** an inevitable approximation error $\epsilon_{app}$ that measures how close the best approximator in the class of linear functions is to the true value function $\mathcal{V}$. We hope it is agreed that the accuracy is fundamentally and unavoidably limited by the power of the approximation class that is being utilized, and one aims to find the best in-class approximator within this given class of functions. It is fundamentally important to characterize how small the error can be made with respect to the best function that can be attained by the function approximation class. To address this question in the heavy-tailed regime, which is frequently encountered in practical applications, we analyze convergence to the best in-class approximator by studying the statistical error in our work. In our case, the heavy-tailed stochastic nature of the reward introduces a severe noise on the gradient with potentially infinite variance of the norm, which we address by a well-designed dynamic truncation mechanism.
> > >
> > > Perhaps the above discussion regarding the approximation error and realizability may be likened to a constrained optimization scenario as an analogy, whereby we are interested in characterizing how close our solution can get to the minimum value of the constrained problem, which has a gap from the unconstrained problem. The capability of the functional class expands the constraint set, and realizability assures that the constraint set encapsulates the unconstrained optimum. In this analogy, the absence of the realizability assumption corresponds to having a constraint set that does not include the unconstrained optimum solution, and therefore introduces an unavoidable gap in the approximation error. We hope the reviewer agrees that: even then, we would like to know how close we can get to this unavoidable level of error with our design.
> > >
> > > **References**
> > >
> > > D. P. Bertsekas, Dynamic Programming and Optimal Control, Vol. II, 3rd Edition, Athena Scientific (2011)
> > >
> > > J. N. Tsitsiklis, and B. Van Roy. "An Analysis of Temporal-Difference Learning with Function Approximation." IEEE Transactions on Automatic Control, 42.5 (1997)

---

> > > > ### Comment · Reviewer_pbb6 · 2023-08-18
> > > >
> > > > I appreciate the authors' response. I agree that without the realizability assumption, there will be an inevitable approximation error. However, when a new estimation algorithm is proposed, readers naturally expect to understand the extent of this approximation error. I hope you agree that if the approximation error turns out to be substantial for any reason, the assurances of sample complexity become less relevant. Take off-policy TD learning as an example, where the basic importance-sampling method might not guarantee convergence in all cases. It is not obvious what unique features your algorithm has that prevent it from encountering significant approximation errors. Perhaps, there are, but this is not shown in the current draft of the work.
> > > >
> > > > I have decided to reduce my score accordingly.

---

> > > > > ### Author Response · Authors · 2023-08-21
> > > > >
> > > > > We are sorry to see that the reviewer has decreased their rating. We believe that there is a misunderstanding, and we are sorry for the confusion. We provide a clarification below to address this.
> > > > >
> > > > > **1. How to achieve small approximation error in our case?** For smaller approximation error, one needs a richer function approximation scheme (please also see the second point in our response below). In our case, this can be achieved by choosing a larger dimension $d$, large radius $\rho > 0$ for the parameter space, and a well-designed set of feature vectors $\{\Phi(x): x\in\mathbb{X}\}$. We first note that our Robust TD learning algorithm with linear function approximation is flexible, and works for *any given* $d, \Phi$ and $\rho > 0$ that defines the linear function approximation class. As such, small approximation errors can be achieved by using a richer function class (large $d$ and $\rho$, and good feature vectors $\Phi$ that leads to full-rank Gram matrix) at the expense of increased sample complexity, which is reflected explicitly in our bounds in **Theorems 1-3**: if we use a larger $\rho$ (which is naturally required if dimension $d$ is large), our error bounds in all cases grow polynomially in $\rho$, which implies larger statistical error to achieve smaller approximation error. This is an instance of the bias-complexity tradeoff: better approximation capability requires larger function classes, making the parameter search harder.
> > > > >
> > > > > For example, in an extreme case where $\mathbb{X}$ is a finite state-space (denoted by integers from 1 to $n$), $d = n$, $\rho = \frac{M}{1-\gamma}$ and $\Phi(x)$ is a unit vector with $1$ at the location of $x$, we obtain the tabular TD learning, and the function approximation error is 0. Of course, the above is tractable only if the state space $\mathbb{X}$ is small, and when $\mathbb{X}$ is large, one needs to use a function approximation scheme with $d \ll |\mathbb{X}|$ for tractability, with the hope that the underlying value function can be well-approximated by some (unknown) parameter with the specified function class. TD learning enters the picture at that point to find this unknown parameter by using only samples from the system. For example, in the case of Linear MDPs (Jin et al., 2020), one may achieve arbitrarily small error by using linear function approximation even when $d \ll |\mathbb{X}|$.
> > > > >
> > > > > We kept our bounds as general as possible, similar to the seminal works (Tsitsiklis and van Roy, 1997; Bhandari et al., 2018), to cover these structured cases with any given $d, \rho, \Phi$.
> > > > >
> > > > > **2. Clarification on approximation error:** The objective of an RL algorithm equipped with function approximation is to achieve the smallest error possible by using the given function approximation architecture. Approximation error is the minimum error that can be achieved by *any* algorithm that uses the given function approximation scheme. We would like to emphasize that the approximation error is independent of the algorithm, and it solely depends how rich the function class is. As such, it is not the task of TD learning to prevent encountering significant approximation errors: TD learning only aims to find the best function within the class defined by the given function approximation architecture. One needs to use a richer function class to avoid large approximation errors at the expense of larger statistical errors (i.e., sample complexity), as we discussed in the previous point.
> > > > >
> > > > > ---
> > > > >
> > > > > Since the main focus of this work is providing the means of managing heavy-tailed noise in the RL framework, we have focused on providing the rigorous findings in the cleanest form. We hope that our response above addresses the questions asked by the reviewer, and we will be happy to include the discussion on how to release the assumption in a revision. Thank you.

---

### Author Rebuttal · Authors · 2023-08-09

We would like to thank the reviewers for their very valuable feedback. In the following, we provide a general response to the questions commonly raised by the reviewers. In order to address specific questions, we will post individual comments.

**1. Realizability assumption and approximation error.** We note that the realizability assumption (Assumption 3) allows us to present our convergence rate results in a clear format. However, all of the results in the paper can be more generally stated without it by incorporating the unavoidable approximation error in the general case. As a showcase, we can state Theorem 1.a without Assumption 3 as follows.

*Theorem 1.a* Let $\epsilon_{app}=\min_{\Theta\in B_2(0,\rho)}\sqrt{\mathbb{E}[|\mathcal{V}(x)-f_\Theta(x)|^2]}$. Under Assumptions 1 and 2.a, with $b_t=(ut)^\frac{1}{1+p}$ and $\eta = \frac{2\rho(1-\gamma)}{(uT)^\frac{1}{1+p}}$, we have $\mathbb{E}|\mathcal{V}(x)-\langle\bar{\Theta}(T),\Phi(x)\rangle|^2\leq \frac{6\rho u^\frac{1}{1+p}}{(1-\gamma)T^\frac{p}{1+p}}+2(\rho+\frac{1}{1-\gamma})\epsilon_{app}$.

Here, note that $\epsilon_{app}=\min_{\Theta\in B_2(0,\rho)}\mathcal{L}(\Theta)$, where $\mathcal{L}(\Theta)=\sqrt{\mathbb{E}|\mathcal{V}(x)-\langle \Theta,\Phi(x)\rangle|^2}$, is the best possible error that can be achieved by using the specified approximation architecture $f_\Theta$. Thus, given $\{\Phi(x)\in\mathbb{R}^d:x\in\mathbb{X}\}$, the goal is to converge to the best $\Theta^\star$ that minimizes the total error $\mathcal{L}(\Theta)$. To make $\epsilon_{app}$ smaller, more powerful approximation schemes must be used. For example, larger dimension $d$ and projection radius $\rho$ implies smaller $\epsilon_{app}$ for linear approximation. The absence of Assumption 3 can be handled exactly the same way for all results in the paper. However, writing all results in this extended form creates an additional notational and conceptual complexity that is not the main focus of this work. As such, we omitted this extension in the original submission. Nevertheless, in the updated version, we will clarify this generalization with an example as we presented above. Similar approximation errors with respect to the true value function $\mathcal{V}$ appears in the seminal works [R1, R2] also, since these works consider convergence to the fixed point of the *projected* Bellman equation. The approximation error is characterized in Theorem 1.(c) in [R1] and Lemma 2 in [R2]. We will clarify this connection with the existing works in our update.

**2. Comparison with existing works.** We would like to thank the reviewers for pointing this out, and we will further clarify our contributions in the paper.

A major challenge that we tackled in this paper is to tame heavy-tailed noise in TD learning that stems from heavy-tailed rewards while keeping the computational efficiency of TD learning as a descent-type iterative stochastic method. In the heavy-tailed regime that we consider, the existing analyses of TD learning do not hold due to infinite variance of the gradient, including the seminal works [R1, R2]. Indeed, in practice, the original TD learning may diverge in the heavy-tailed regime, as we demonstrated in our numerical investigations.
There are many choices of robust mean-estimation techniques for heavy-tailed random variables, including median-of-means, Catoni’s M-estimator. In our design, we adopted the truncated (i.e., clipped) mean estimator to preserve the computational efficiency of TD learning as a descent-type iterative algorithm while taming heavy-tailed phenomenon in RL. However, achieving this goal introduces the following two main critical challenges, which we resolved in our work:
- The use of clipping introduces a bias in the stochastic semi-gradient in TD learning, and we identified in our Lyapunov analysis that this bias needs to be controlled along with the variance by adjusting the clipping radii appropriately. In particular, we achieved a favorable balance between this bias and variance of the stochastic semi-gradient by developing problem-dependent sequences of clipping radii, which drive the statistical error to 0 at the rates specified in the paper.
- The original truncated mean estimator is used for estimating the mean of a random variable from iid samples, which is not valid in TD learning. To address this non-iid behavior, we devised new Martingale techniques based on Freedman’s inequality on the Lyapunov drift in our analysis of Robust TD learning.

We acknowledge that these contributions were not clearly articulated in the original submission. Thanks to the comments of the reviewers, we will make sure that they are clarified in the updated version of the paper.

**3. Heavy-tailed regime** Our setting encapsulates reward distributions with a finite mean but potentially infinite variance as specified by our Assumption 1. This class includes well-known heavy-tailed distributions such as *Pareto* distribution with parameter $\alpha \in (1,2]$, but excludes more extreme distributions with infinite mean such as Cauchy, which would make the total discounted reward $\mathcal{V}$ infinite. We would like to note that, in many important RL applications, the reward distribution satisfies Assumption 1. For instance, in scheduling and routing applications in computing systems, job sizes and service times are heavy-tailed with finite mean and infinite variance: $p\in(0,0.25)$ in Bellcore, $p\in (0, 0.1)$ for the Web [R3] with $p$ in our Assumption 1. We will be happy to clarify these points in our update.

[R1] J. Tsitsiklis and B. van Roy. ”Analysis of temporal-diffference learning with function approximation.” Advances in neural information processing systems 9 (1996)

[R2] J. Bhandari et al., ”A Finite Time Analysis of Temporal Difference Learning With Linear Function Approximation.” arXiv preprint arXiv:1806.02450 (2018)

[R3] M. Harchol-Balter, ”Task Assignment with Unknown Duration.” Journal of the ACM 49.2:260-288 (2002)

---

### Decision · Program_Chairs · 2023-09-21

**Decision:**

Accept (poster)

**Comment:**

This work explores the limitations of classical TD learning when dealing with heavy-tailed rewards. To address this, the authors introduce a novel technique called robust TD learning, which features a dynamic gradient clipping mechanism. Theoretical convergence rates for both expected and high-probability outcomes are provided to ensure the new algorithm's effectiveness under heavy-tailed rewards. Empirical experiments are conducted to validate the theoretical findings.

Evaluation:
The paper is clearly written and easy to comprehend.

While the research problem is compelling, some reviewers feel that the theoretical contributions lack depth. The authors' rebuttal did not fully convince the reviewers, and the meta-reviewer partially agrees with this assessment.

The authors do sufficiently address the issue of approximation error in their response. However, the RL setting described in the paper could be more nuanced. For example, is ergodicity required? How does the policy optimization approach solve the problem of online exploration?

In summary, the paper is on the borderline of acceptance.